# Structure and dynamics of endogenous cardiac troponin complex in human heart tissue captured by native nanoproteomics

Emily A. Chapman [1,7], David S. Roberts [1,7], Timothy N. Tiambeng[1], Jãán Andrews [1], Man-Di Wang[1], Emily A. Reasoner[1], Jake A. Melby [1], Brad H. Li[2], Donguk Kim[3], Andrew J. Alpert[4], Song Jin [1] ✉ & Ying Ge [1,5,6] ✉

Protein complexes are highly dynamic entities that display substantial diversity in their assembly, post-translational modifications, and non-covalent interactions, allowing them to play critical roles in various biological processes. The heterogeneity, dynamic nature, and low abundance of protein complexes in their native states present challenges to study using conventional structural biology techniques. Here we develop a native nanoproteomics strategy for the enrichment and subsequent native top-down mass spectrometry (nTDMS) analysis of endogenous cardiac troponin (cTn) complex directly from human heart tissue. The cTn complex is enriched and purified using peptide-functionalized superparamagnetic nanoparticles under non-denaturing conditions to enable the isotopic resolution of cTn complex, revealing their complex structure and assembly. Moreover, nTDMS elucidates the stoichiometry and composition of the cTn complex, localizes $Ca^{2+}$ binding domains, defines cTn-$Ca^{2+}$ binding dynamics, and provides high-resolution mapping of the proteoform landscape. This native nanoproteomics strategy opens a paradigm for structural characterization of endogenous native protein complexes.

The vast majority of proteins within a cell assemble into protein complexes to perform specific functions and play crucial roles in regulating diverse biological processes[1,2]. Thus, comprehensive characterization of the structure and dynamics of endogenous protein complexes is essential for understanding fundamental biological processes and disease mechanisms to develop new therapeutic interventions[1,2]. Protein complexes are highly dynamic entities with substantial diversity in their assembly, post-translational modifications (PTMs), and non-covalent interactions. Moreover, endogenous protein complexes often exist in low abundance in their native states[2]. These present tremendous challenges to study their

structure and dynamics using traditional structural biology techniques such as X-ray crystallography, nuclear magnetic resonance (NMR) spectroscopy, and cryo-electron microscopy (cryo-EM)[2]. Native top-down mass spectrometry (nTDMS), a technique combining native MS[3–7] and top-down proteomics[8–13], has emerged as a powerful structural biology tool for the characterization of protein complexes[14–19]. In nTDMS, intact proteins are introduced into the mass spectrometer under non-denaturing conditions, preserving the non-covalent interactions between protein subunits and their associated ligands as well as PTMs. The intact proteins are then fragmented in the gas phase to map the PTMs and ligand binding sites[14].

[1]Department of Chemistry, University of Wisconsin-Madison, Madison, WI 53706, USA. [2]Department of Biochemistry, University of Wisconsin-Madison, Madison, WI 53706, USA. [3]Department of Chemical and Biological Engineering, University of Wisconsin-Madison, Madison, WI 53706, USA. [4]PolyLC Inc, Columbia, MD 21045, USA. [5]Department of Cell and Regenerative Biology, University of Wisconsin-Madison, Madison, WI 53705, USA. [6]Human Proteomics Program, School of Medicine and Public Health, University of Wisconsin-Madison, Madison, WI 53705, USA. [7]These authors contributed equally: Emily A. Chapman, David S. Roberts. ✉e-mail: jin@chem.wisc.edu; ying.ge@wisc.edu

nTDMS enables the structural characterization of macromolecular protein complexes, subunit stoichiometry, non-covalent interactions, as well as the analysis of their proteoforms—the diverse protein products arising from alternative splice isoforms, genetic variations, and PTMs[9,20,21]. However, so far only over-expressed recombinant or high-abundance proteins and protein complexes have been characterized by nTDMS. Significant challenges remain in the structural characterization of endogenous protein complexes that are present in relatively lower-abundance due to the difficulty in isolating them and sensitivity required to resolve heterogenous complexes[2].

Here, we have developed a native nanoproteomics platform integrating the native enrichment of protein complexes directly from tissues using surface-functionalized superparamagnetic nanoparticles (NPs) with high-resolution nTDMS to characterize the structure and dynamics of endogenous protein complexes. Specifically, we applied this method to enrich and structurally elucidate the heterotrimeric cardiac troponin (cTn) complex (-77 kDa), a key protein complex essential to cardiac muscle contraction, directly from human heart tissues. The cTn complex is a master regulator of cardiac contraction and represents the $Ca^{2+}$ sensitive switch[22,23] of striated muscles assembled from three molecular subunits: troponin C (TnC), the $Ca^{2+}$-binding subunit; cardiac troponin I (cTnI), the actin-myosin inhibitory subunit; and cardiac troponin T (cTnT), the thin-filament anchoring subunit[24,25]. Both cTnI and cTnT serve as gold standard biomarkers for diagnosing acute coronary syndrome due to their cardiac specificity and their release into the bloodstream following cardiac injury[26]. Moreover, the association of $Ca^{2+}$ ions with the TnC subunit along with phosphorylation of the cTnI subunit together initiate a cascade of molecular events on the thin filament and induce actin-myosin cross-bridge formation necessary for cardiac contraction[27,28]. However, only partial structural information of the core domain of the human cTn complex in its $Ca^{2+}$ saturated state has been obtained from conventional X-ray crystallography excluding the intrinsically disordered but functionally critical regions of the N- and C- terminal regions of both cTnI and cTnT[29]. Moreover, the cTn structure is highly dynamic due to $Ca^{2+}$ binding[25,30,31] and PTMs[11,32,33] that regulates muscle contraction, yet traditional methods have not effectively captured these dynamic conformational changes, primarily because they are limited to visualizing the static state of the thin filament[34]. Furthermore, recombinantly expressed cardiac troponin subunits are frequently utilized and incorporated into reconstituted cTns or cardiac thin filaments to investigate the structure-function relationships of the complex as well as $Ca^{2+}$ sensitivity[35–37]. However, the lack of post-translational machinery in prokaryotic cell expression systems presents intrinsic limitations when recapitulating eukaryotic, post-translationally modified endogenous proteins such as phosphorylated cTn subunits, using recombinant protein systems[38,39].

In this work, we enrich and purify cTn complex directly from human heart tissue and achieve isotopic resolution of endogenous cTn complex to reveal cTn complex structure and assembly. Our results elucidate the stoichiometry and composition of the heterotrimeric cTn complex, define the conformational roles of cTn·$Ca^{2+}$ binding dynamics, locate the $Ca^{2+}$ binding domains (II-IV), and map the proteoform landscape with direct analysis of the stoichiometry of various proteoforms. Overall, this work demonstrates that nTDMS, enabled by native nanoproteomics, can comprehensively characterize the structure and dynamics of endogenous protein complexes.

## Results

### A native nanoproteomics platform for the enrichment and comprehensive characterization of endogenous protein complexes

Our antibody-free native nanoproteomics platform integrates native enrichment and purification of endogenous protein complexes using peptide-functionalized superparamagnetic iron oxide (magnetite,

$Fe_3O_4$) NPs (NP-Pep) followed by comprehensive characterization using nTDMS (Fig. 1). Building on our previous denatured NP enrichment study[40], we sought to establish a method that could enrich and purify native protein complexes directly from human heart tissue. We first optimized the native protein extraction buffers to effectively extract intact protein complexes from human heart tissue using a high ionic strength lithium chloride (LiCl) buffer at physiological pH and a phosphate wash buffer for multiple depletions of highly abundant cytosolic proteins (Fig. 1a, Supplementary Table 1). Next, we hypothesize that the specifically designed peptide[41] on the NP surface which contains a combination of charged and aromatic residues that leads to the specific binding to the protein complex of interest, would be amenable to a competitive elution strategy using amino acids[42–44]. L-Arg and L-Glu are amino acids previously known to improve protein solubility and in-solution stability by forming protonated clusters[43], and were included to enhance protein complex elution efficacy. Moreover, imidazole, a positively charged aromatic small molecule which functions as the side chain of histidine[42], was added to disrupt the polar interactions of the peptide functionalized on the NP surface for the native competitive elution of NP-captured protein complexes, without disrupting the intermolecular interactions between the protein subunits. We found that the optimal buffer composition for effective native competitive elution of NP-captured protein complexes was a combination of 750 mM L-Arg, 750 mM imidazole, and 50 mM L-Glu (pH 7.5) as native elution efficiency of amino acid solutions increases with their concentration[45] (Fig. 1a, Supplementary Fig. 1).

For a native proteomics workflow, the NP-Pep was incubated with sarcomeric protein mixtures, magnetically isolated to remove non-specifically bound proteins, and the bound protein complexes were eluted off the NP-Pep using the above optimized native elution buffer cocktail. Non-MS compatible buffers and salts were removed using either online or offline size-exclusion chromatography (SEC) to transfer the complexes into a MS-compatible ammonium acetate solution. Subsequently, the enriched protein complexes were subjected to various nTDMS techniques to characterize protein complex structure, assembly, and dynamics including: online SEC for rapid screening of protein complexes (Fig. 1b), ultrahigh-resolution Fourier transform ion cyclotron resonance (FTICR)-MS to characterize protein complex stoichiometry and proteoform landscape (Fig. 1c), and trapped ion mobility (TIMS)-MS to resolve the structural dynamics of non-covalent interactions in regulating protein complex conformational change (Fig. 1d). These techniques allowed for direct analysis of structural features revealing the molecular composition, stoichiometry, and non-covalent interaction of endogenous protein complexes (Fig. 1e–f).

### Native enrichment of endogenous cTn complex from human heart tissue

To enrich cTn complex from human heart tissues, the $Fe_3O_4$ NPs were functionalized with a 13-mer peptide (NP-Pep; HWQIAYNEHQWQC)[40,41] with high binding affinity ($K_d$ = 270 pM) towards cTnI under non-denaturing condition. Native cTn complex enrichment yielded TnC, cTnI, and cTnT in approximately a 1:1:1 ratio (Supplementary Fig. 2, Supplementary Fig. 3a), reflecting the composition of the heterotrimeric cTn complex in the sarcomere. Moreover, cTn could be enriched directly from human heart tissue across multiple NP-Pep synthetic batches and replicates (Supplementary Fig. 2). To demonstrate the enrichment and purification of all three cTn subunits while preserving their proteoforms, we further used an online reverse-phase liquid chromatography (RPLC) top-down tandem MS (MS/MS) method comparing the initial sarcomere protein loading mixture (L), the resulting flow through (F), and the final elution mixtures (E) (Supplementary Fig. 3b). Not only were all three subunits of the cTn complex, cTnI, cTnT and TnC, significantly enriched, but also the PTM profiles of endogenous cTnI, cTnT, and TnC were faithfully preserved, without

introducing artifactual modifications (Supplementary Fig. 4). The top-down proteomics results provide a bird's eye view of the proteoform landscape of TnC, cTnT, and cTnI for direct analysis of the stoichiometry of their various proteoforms. Moreover, this native NP enrichment strategy was found to be highly reproducible across three independent donor heart tissues (Supplementary Fig. 5).

We next investigated whether the enriched endogenous cTn complex could be analyzed under native conditions by SEC-MS using an online buffer exchange (OBE) method[46,47] for rapid analysis of native protein complexes after separation from MS-incompatible buffers (Supplementary Fig. 6). Upon analysis of the resulting SEC-MS chromatograms, protein complexes were effectively separated from nonvolatile buffer components within 7 minutes (Supplementary Fig. 6). Additionally, heterotrimeric cTn complex ($z$ = 16+-21+) were reproducibly enriched across three independent elution mixtures from the same heart tissue sample highlighting the reproducibility of this SEC-OBE native MS method (Supplementary Fig. 7).

### Structural heterogeneity of endogenous cTn complex revealed by nTDMS

To comprehensively characterize the endogenous heterotrimeric cTn complex, we employed a nTDMS approach using an ultrahigh-resolution FTICR mass spectrometer for unequivocal proteoform sequencing and protein complex characterization. Native mass spectra of the enriched cTn complex revealed a charge state distribution of 18+ to 21+ (3800 $m/z$ to 4300 $m/z$) with the most abundant charge state ($z$ = 19+) detected between 4050 $m/z$ to 4080 $m/z$ (Fig. 2, Supplementary

Fig. 8). In-depth examination of the endogenous cTn complex revealed four unique complex proteoforms comprised of both covalent and non-covalent modifications (Fig. 2b). All heterotrimer cTn complex proteoforms were identified with high mass accuracy (<2 ppm) (Fig. 2b, Supplementary Table 2). Significantly, the predominant forms of the heterotrimeric cTn complex were revealed to comprise of mono-phosphorylated cTnT, mono-phosphorylated and bis-phosphorylated cTnI, and TnC with three bound $Ca^{2+}$ ions (most abundant cTn complex MW = 77136 Da). These results show that the human cTn complex exists in structurally diverse molecular compositions in the sarcomere with highly heterogenous covalent and non-covalent modifications that yield a suite of different intact assemblies.

### Sequence-specific structural elucidation of cTn complex quaternary structure through nTDMS

Next, we performed complex-up native MS analysis[48] with collisionally activated dissociation (CAD)[49] to elucidate the stoichiometry and composition of the heterotrimeric cTn complex (Fig. 3). First, the intact heterotrimeric cTn complex ($z$ = 18+-20+) was isolated with no dissociated cTn subunits present in the resulting mass spectra (Fig. 3a). Subunit ejection of the cTn complex was then observed by CAD (Fig. 3b). We detected the intact cTn(I-C) dimer at 2700 $m/z$ to 3600 $m/z$ ($z$ = 12+-16+, MW = 42556 Da), cTnT monomer at 2500 $m/z$ to 3200 $m/z$ ($z$ = 11+-14+, MW = 34580 Da), and TnC monomer at 2600 $m/z$ to 3100 $m/z$ ($z$ = 6+-7+, MW = 18520 Da). Ejected cTnI monomer was not detected in appreciable abundance. We suspect that this may be due to maintaining native conditions during our MS analysis which seems to

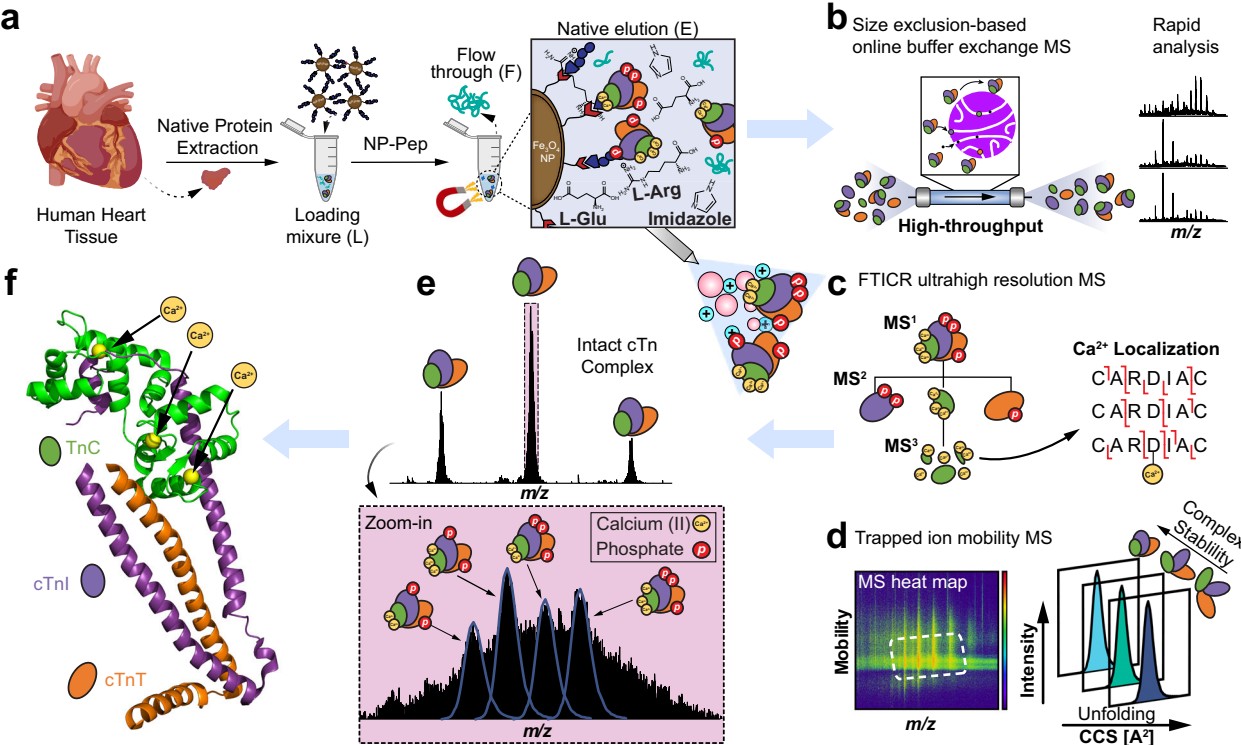

**Fig. 1 | Native nanoproteomics platform for the characterization of endogenous cTn complex from human heart tissue. a** Sarcomeric proteins were extracted using a high ionic strength lithium chloride (LiCl) buffer at physiological pH. The heart tissue extract (loading mixture, L) is then incubated with peptide-functionalized nanoparticles (NP-Pep). Following magnetic isolation, the non-specifically bound proteins are removed as flow through (F). The bound protein complexes are then eluted (E) off the NPs using a native elution buffer containing L-glutamic acid, L-arginine, and imidazole. Native top-down MS (nTDMS) analysis of enriched protein complexes proceeds by either (**b**) native size exclusion chromatography (SEC) for online buffer exchange and rapid analysis, (**c**) ultrahigh-resolution Fourier transform ion cyclotron (FTICR)-tandem MS (MS/MS) analysis for probing complex heterogeneity, stoichiometry, and localization of Ca(II) ions, or (**d**) trapped ion mobility spectrometry (TIMS) coupled with MS (timsTOF) for structural characterization of complex-Ca(II) binding dynamics. **e** High-resolution mass spectra of endogenous cardiac troponin (cTn) heterotrimeric complexes enriched by NP-Pep directly from human heart tissue and analyzed by nTDMS. **f** Structural representation showing the cTn heterotrimeric complex. Troponin C (TnC) is depicted in green, cardiac troponin I (cTnI) in purple, cardiac troponin T (cTnT) in orange, and Ca(II) ions in yellow. PDB: 1J1E.

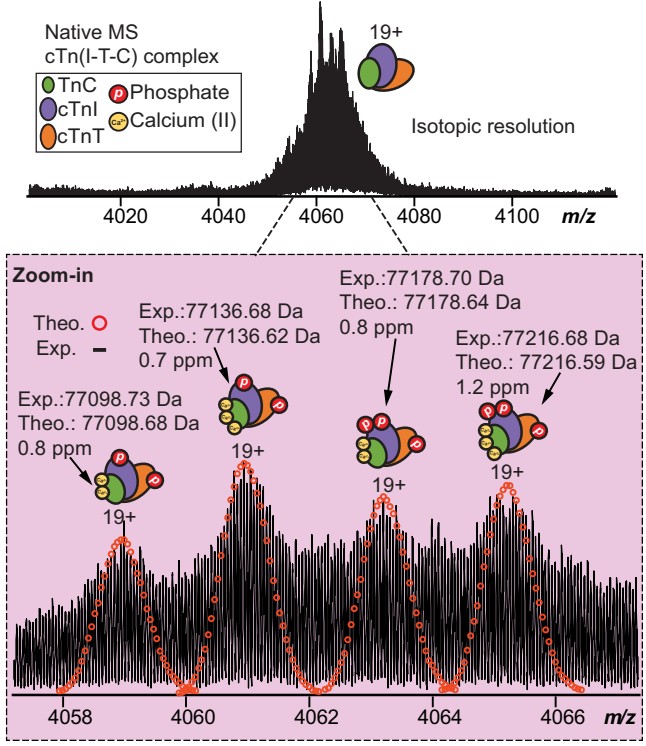

**Fig. 2 | High-resolution native top-down MS analysis demonstrates heterogeneity of endogenous heterotrimeric cTn complex.** Native MS spectra of the endogenous cTn complex using ultrahigh-resolution FTICR-MS to achieve isotopic resolution. Most abundant charge state ($z$ = 19+) is isolated. Zoomed-in mass spectra shows the isotopically resolved cTn complex proteoforms detected. Theoretical isotope distributions (red circles) are overlaid on the experimentally obtained mass spectrum to illustrate the high mass accuracy. All individual ion assignments are within 2 ppm of the theoretical mass.

preserve the strong binding affinity of cTnI to TnC[50,51]. Additionally, it is possible that the ejected cTnI monomer may appear at too low of a charge (i.e. too high of $m/z$ range) for us to meaningfully detect during our complex-up native MS analysis.

To gain additional structural and sequence-specific information of the endogenous cTn complex, we performed complex-down analysis[48] of the dissociated cTn subunits (Fig. 3c, d). The isolation of ejected cTnT monomers ($z$ = 14+) revealed multiple proteoforms, including unphosphorylated cTnT, monophosphorylated cTnT ($p$cTnT), and phosphorylated cTnT with C-terminal Lys truncation ($p$cTnT [aa 1-286]) (Fig. 3c). Further CAD fragmentation revealed fragments which suggest that the $C$-terminus of cTnT is more solvent exposed than the $N$-terminus forming the heterotrimer interface (Fig. 3c, Supplementary Fig. 9). Next, the isolation of ejected cTn(I-C) dimer ($z$ = 14+) revealed six unique proteoforms with different $Ca^{2+}$ occupancy and phosphorylation states: unphosphorylated cTnI associated with TnC and 2 $Ca^{2+}$, unphosphorylated cTnI associated with TnC and 3 $Ca^{2+}$, monophosphorylated cTnI ($p$cTnI) associated with TnC and 2 $Ca^{2+}$, $p$cTnI associated with TnC and 3 $Ca^{2+}$, bisphosphorylated cTnI ($pp$cTnI) associated with TnC and 2 $Ca^{2+}$, and $pp$cTnI associated with TnC and 3 $Ca^{2+}$ (Fig. 3d). The cTn(I-C) dimer precursor ions ($z$ = 14+) were further subjected to CAD fragmentation requiring high collision voltage (70 V) to obtain additional structural information (Fig. 3d, Supplementary Fig. 10). We observed phosphorylation of dissociated cTnI monomer at Ser22 and Ser23, which are the targets of PKA-mediated phosphorylation[32]. Our nTDMS analysis also suggests that both the intrinsically disordered $C$- and $N$-termini of cTnI are more solvent exposed than the stable internal regions[35] that form the subunit-subunit interfaces of the cTn complex. While the relative stability of

the different cTn molecular forms could be inferred based on how they responded to the nTDMS analysis, we could not infer exactly how $Ca^{2+}$ binding and phosphorylation impacted the stability of the cTn complex. Moreover, we did not observe lowly charged "stripped" cTn heterotrimer complexes that are expected at a much higher $m/z$ likely due to the limitation of our FTICR mass spectrometer which is not optimized for the detection of ions at very high $m/z$[52].

## Direct localization of $Ca^{2+}$ binding domains in the endogenous cTn complex

We have characterized the three $Ca^{2+}$ binding domains present in endogenous TnC using nTDMS (Fig. 4). TnC is an EF-hand $Ca^{2+}$-binding protein that is an essential $Ca^{2+}$ sensing molecular subunit in the heterotrimeric cTn complex[30,31]. Human TnC consists of three functional metal-binding motifs (domains II–IV) that can be occupied by $Ca^{2+}$ and are responsible for regulating cardiac muscle contraction[39]. $Ca^{2+}$ binds to domain II with low-affinity (-$10^{-5}$ M) and serves as the "regulatory" domain for initiating cardiac contraction[53]. In contrast, domains III and IV, referred to as "structural" sites, bind $Ca^{2+}$ with high-affinity (-$10^{7}$ $M^{-1}$)[54]. Domains III and IV consistently remain saturated with $Ca^{2+}$ during relaxation and contraction, whereas Domain II is only occupied during contraction[55]. Notably, domains III and IV exhibit a slower $Ca^{2+}$ exchange rate than domain II[54,56]. The nTDMS isolation spectra of ejected TnC monomer ($z$ = 8+) revealed multiple proteoforms of endogenous TnC in its $Ca^{2+}$-bound states (Fig. 4a). Specifically, we observed TnC with 0, 1, 2, and 3 $Ca^{2+}$ ions bound with baseline isotopic resolution and high mass accuracy. The relative proportion of singly, doubly, and triply bound $Ca^{2+}$ states was found to be approximately 0.2, 0.5, and 0.1, respectively (Supplementary Fig. 11).

The essential amino acid regions necessary for an individual $Ca^{2+}$ ion to associate with TnC domains were next determined through nTDMS analysis. Aspartic acid and glutamic acid have the highest binding affinities for $Ca^{2+}$ at neutral pH[39,57]. To localize $Ca^{2+}$ binding domains, TnC proteoforms were subjected to CAD to generate extensive backbone fragmentation through collisional activation ramping (Fig. 4b, c, Supplementary Fig. 12). We found TnC domain III to be the least vulnerable $Ca^{2+}$ binding region to collisional activation, while domain II was found to be the most vulnerable $Ca^{2+}$ binding region. To localize the primary region for $Ca^{2+}$ binding that is least vulnerable to collisional activation, we isolated the TnC proteoform at 2312 $m/z$ and performed CAD to yield product ions $y_{52}$ + $Ca^{2+}$, $y_{30}$, $b_{115}$ + $Ca^{2+}$, and $b_{109}$ (Supplementary Fig. 13). Therefore, the primary $Ca^{2+}$ binding domain was localized to [113]DLD[115] in domain III. The next $Ca^{2+}$ binding domain was localized to the structural region between [141]DKNND[145] in domain IV by first isolating the TnC proteoform at 2316 $m/z$ and then generating CAD product ions $b_{140}$ + $Ca^{2+}$, $b_{145}$ + 2 $Ca^{2+}$, $y_{16}$, and $y_{22}$ + $Ca^{2+}$ (Supplementary Fig. 14). Finally, the most vulnerable $Ca^{2+}$ binding region was localized to regulatory domain II between [73]DFDE[76] by isolating the TnC proteoform at 2321 $m/z$ and generating CAD product ions $b_{65}$, $b_{91}$ + $Ca^{2+}$, $y_{85}$ + 2 $Ca^{2+}$, and $y_{94}$ + 3 $Ca^{2+}$ (Supplementary Fig. 15). This study localizes endogenous TnC $Ca^{2+}$ binding regions to domains II, III, and IV in the cTn complex (Fig. 4d). Additionally, a molecular depiction of the divalent association of $Ca^{2+}$ to amino acid residues [73]D and [76]E in TnC domain II is illustrated in Fig. 4e.

## Determination of cTn-$Ca^{2+}$ binding and conformational dynamics

The binding of $Ca^{2+}$ to TnC has extensive effects on the heterotrimeric cTn complex function and structure in regulating cardiac contraction[25,29,31,58]. The core of the heterotrimeric cTn complex maintains a stable conformation, while flexible regions undergo extensive conformational changes when $Ca^{2+}$ binds to TnC[34]. This transition shifts the cTn complex from a 'closed' state where muscle contraction is prevented by cTnI binding to actin, to an active 'open state'. In this active state, the N-terminal domain of TnC opens,

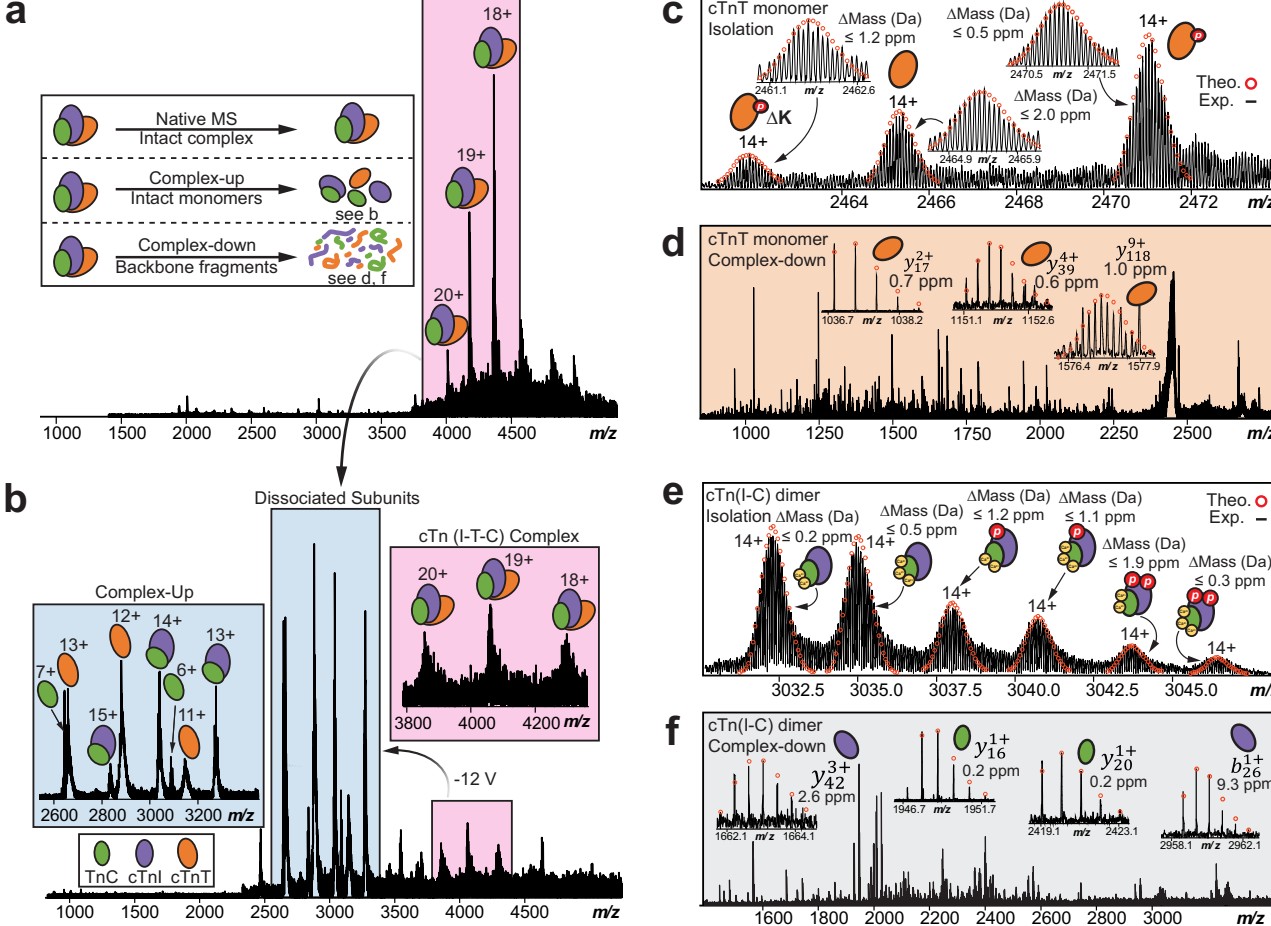

**Fig. 3 | Native top-down MS (nTDMS) analysis for the characterization of cTn complex directly from human heart tissue. a** Isolation of the heterotrimeric cTn complex achieved by ultrahigh-resolution FTICR-MS with charge states labeled. Inset shows three common approaches used for nTDMS analysis. **b** Complex-up analysis of the cTn complex ejects cTn monomers and dimer to reveal hetero-trimeric complex stoichiometry. **c** Isolation of ejected cTnT native monomer ($z$ = 14+) followed by complex-down analysis using collisionally activated dissociation (CAD) fragmentation. Theoretical isotope distributions (red circles) are overlaid on the experimentally obtained mass spectrum to illustrate the high mass accuracy. All individual ion assignments are < 2 ppm from the theoretical mass. **d** Representative MS/MS spectra and CAD fragment ions ($y_{17}^{2+}$, $y_{39}^{4+}$, $y_{118}^{9+}$) obtained from the nTDMS analysis. **e** Isolation of ejected cTn native dimer ($z$ = 14+) followed by complex-down analysis using CAD fragmentation. Theoretical isotope distributions (red circles) are overlaid on the experimentally obtained mass spectrum to illustrate the high mass accuracy. All individual ion assignments are within 2 ppm of the theoretical mass. **f** Representative MS/MS spectra and CAD fragment ions (cTnI: $y_{42}^{3+}$, $b_{26}^{1+}$, TnC: $y_{16}^{1+}$, $y_{20}^{1+}$) obtained from the nTDMS analysis.

allowing for the binding of cTnI, and facilitating the interaction between actin and myosin, ultimately leading to cardiac muscle contraction. Due to the conformational heterogeneity and presence of intrinsically disordered regions along the heterotrimeric cTn complex, it is challenging to obtain crystal structures of cTn in its active and closed states upon $Ca^{2+}$ binding using traditional structural biology techniques[29]. Therefore, to assess the intricate relationship between cTn-$Ca^{2+}$ binding and conformational dynamics, we performed native TIMS-MS analysis. TIMS positions ions in an electrical field against a moving buffer gas to determine ion mobility ($1/K_0$) values for structural analysis of native proteins and protein complexes[12,59,60]. The measured mobilities are converted into collision cross-section (CCS) values to provide structural information on ion shape and size to evaluate conformational changes in the protein's three-dimensional structure[61,62].

First, the native TIMS-MS parameters were optimized using bovine serum albumin (BSA, ~66 kDa) due to BSA having similar $m/z$ and ion mobility regions as the cTn complex. Modifying the desolvation parameters proved critical for effective high-resolution ion mobility separation of BSA conformers (Supplementary Fig. 16). We next used TIMS-MS to separate and analyze the endogenous TnC

monomer and cTn(I-C) dimer $Ca^{2+}$-bound proteoforms (Fig. 5). The CCS values for TnC monomer with 0, 1, 2, and 3 $Ca^{2+}$ ions bound were determined to be 1853, 1849, 1829, and 1844 $Å^2$, respectively (Fig. 5a, b). The experimentally obtained CCS values revealed a more compact conformation for TnC than was predicted by IMPACT[63] calculation derived from available TnC crystal structures (Supplementary Table 3). On the other hand, the CCS values for cTn(I-C) dimer with 2 (3623 $Å^2$) and 3 (3640 $Å^2$) $Ca^{2+}$ ions were in excellent agreement with the calculated IMPACT CCS values (Fig. 5c-d, Supplementary Table 3). For TnC monomer, the structure remains in an open conformation in its apo and 1 $Ca^{2+}$ bound states, whereas TnC shifts to a closed state upon 2 $Ca^{2+}$ ions binding (Fig. 5e). Interestingly, TnC shows a slightly more open conformation upon 3 $Ca^{2+}$ ions binding. We reason that TnC with 2 $Ca^{2+}$ ions bound forms a more compact conformation because $Ca^{2+}$ structural binding domains III and IV are typically saturated at resting diastolic concentrations, but when the third $Ca^{2+}$ ion binds to domain II, TnC adopts a more open conformation to prepare the N-terminal region of TnC for binding to the C-terminal region of cTnI to initiate cardiac muscle contraction[31,64]. Likewise, binding of three $Ca^{2+}$ ions to cTn(I-C) dimer shifts the dimer to a more open conformation when compared to the 2 $Ca^{2+}$ bound state (Fig. 5f). These results

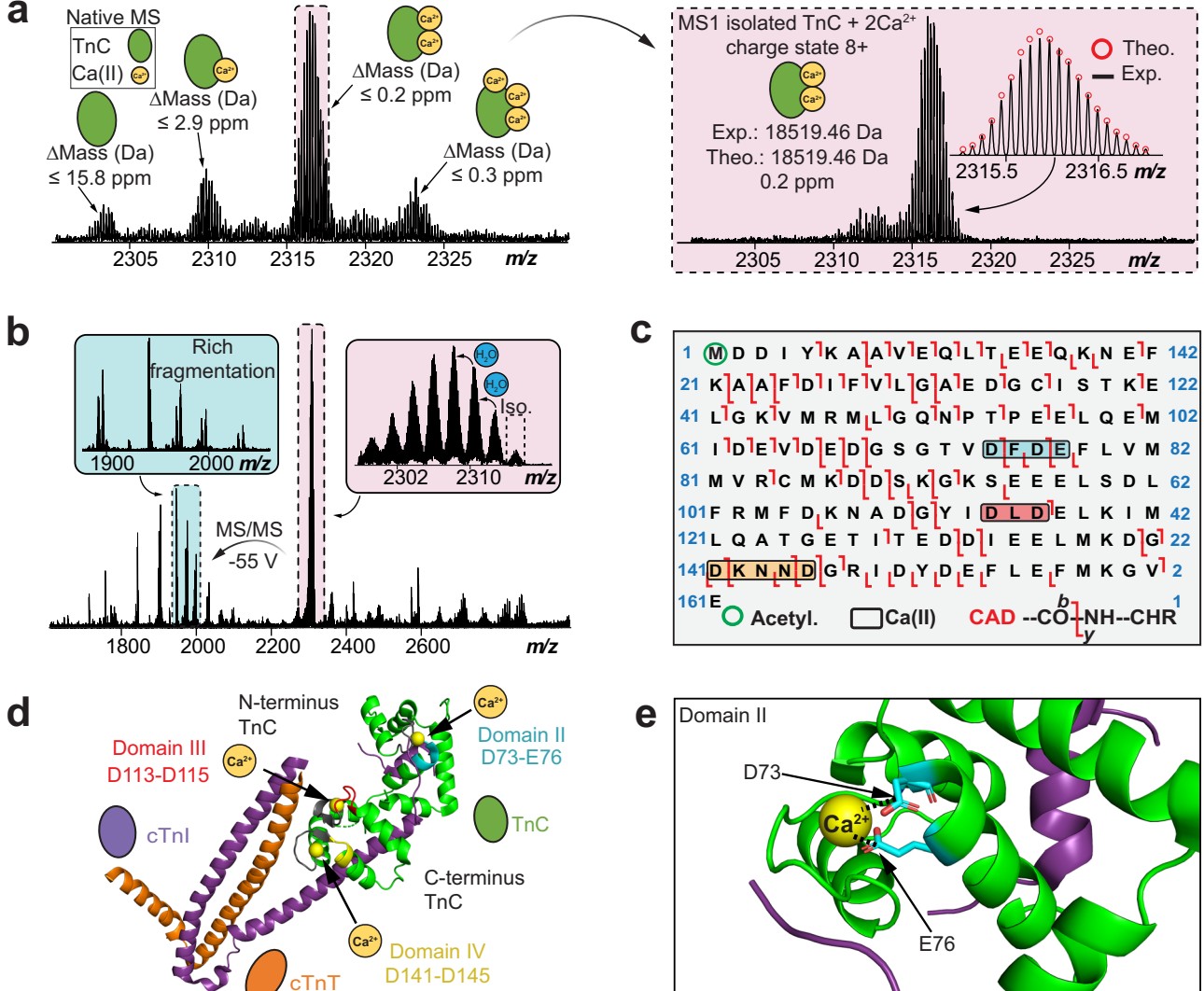

**Fig. 4 | Native top-down MS (nTDMS) enables localization of endogenous Ca(II)-binding domains within TnC. a** Isolation of native TnC monomer ($z$ = 8+) showing proteoforms with differential Ca(II) binding using ultrahigh-resolution FTICR-MS. (Inset) Isolation of TnC + 2 Ca(II) proteoform ($z$ = 8+). Theoretical isotope distributions (red circles) are overlaid on the experimentally obtained mass spectrum to illustrate the high mass accuracy proteoform characterization. All individual ion assignments are within 16 ppm from the theoretical mass. **b** MS/MS characterization of TnC + 2 Ca(II) proteoform isolated from the quadrupole window centered at 2316 $m/z$ by collisionally activated dissociation (CAD) fragmentation. **c** nTDMS CAD

fragmentation map showing the localization of three Ca(II) ions and N-terminal acetylation in TnC monomer achieving ~63% total bond cleavage. We observed sequential binding of Ca(II) ions to TnC with Ca(II) first binding to domain III (red), then domain IV (yellow), and lastly domain II (blue). **d** Structural representation of the cTn complex with the experimentally defined Ca(II)-binding domains (II–IV) highlighted (domain II, blue; domain III, red; domain IV, yellow, UniprotKB annotations, gray). TnC is depicted in green, cTnI in purple, cTnT in orange, and Ca(II) ions in yellow. PDB: 1J1E. **e** Magnified view of TnC Ca(II)-binding in domain II showing possible coordination with amino acid residues D73 and E77. PDB: 1J1E.

suggest that cTnI and TnC form a more open conformation when saturated with Ca²⁺ ions in preparation for cTnT engagement along the thin filament for subsequent cTn complex formation and cardiac muscle contraction[65].

The structural roles of Ca²⁺ in maintaining the intact heterotrimer stability were further profiled by native TIMS-MS analysis (Fig. 6). We added EGTA to sequester Ca²⁺ from the intact heterotrimer complex and probe the specific contribution of individual Ca²⁺ binding regions to cTn complex stability (Fig. 6a). First, the heterotrimeric cTn complex ($z$ = 18+ to 21+) was resolved by native TIMS-MS analysis (Fig. 6b, c) without EGTA incubation. The CCS of the native cTn complex with all Ca²⁺ occupied was determined to be 4880 Å² which was comparable to the CCS value calculated using the IMPACT[63] method (4192 Å²) and is in good agreement with the previously reported partial crystal structure[29] that is truncated due to intrinsically disordered regions present within the cTn complex

(Supplementary Table 3). Under increasing concentrations of EGTA (25, 50, and 100 mM) to sequester cTn-bound Ca²⁺, the intact cTn complex was gradually unfolded, as evidenced by an increase in the average charge stage as well as a shift in the charge state envelope to lower $m/z$ regions for the cTn complex. TIMS analysis of the cTn complex revealed protein conformer enhancement and loss of protein complex stability which correlated with increasing EGTA concentration (Fig. 6d–f, Supplementary Table 4). Additionally, plotting the experimentally determined CCS values as a function of charge state for TnC monomer, cTn(I-C) dimer, and cTn(I-T-C) heterotrimer illustrates the linear relationship between charge state and native protein conformation for each cTn molecular form (Supplementary Fig. 17). While TIMS analysis offers comprehensive insights into protein complex size and stability, determining the precise extent of Ca²⁺ binding within the cTn complex remains challenging, as only an average mass for the ensemble can be attained. We suspect that,

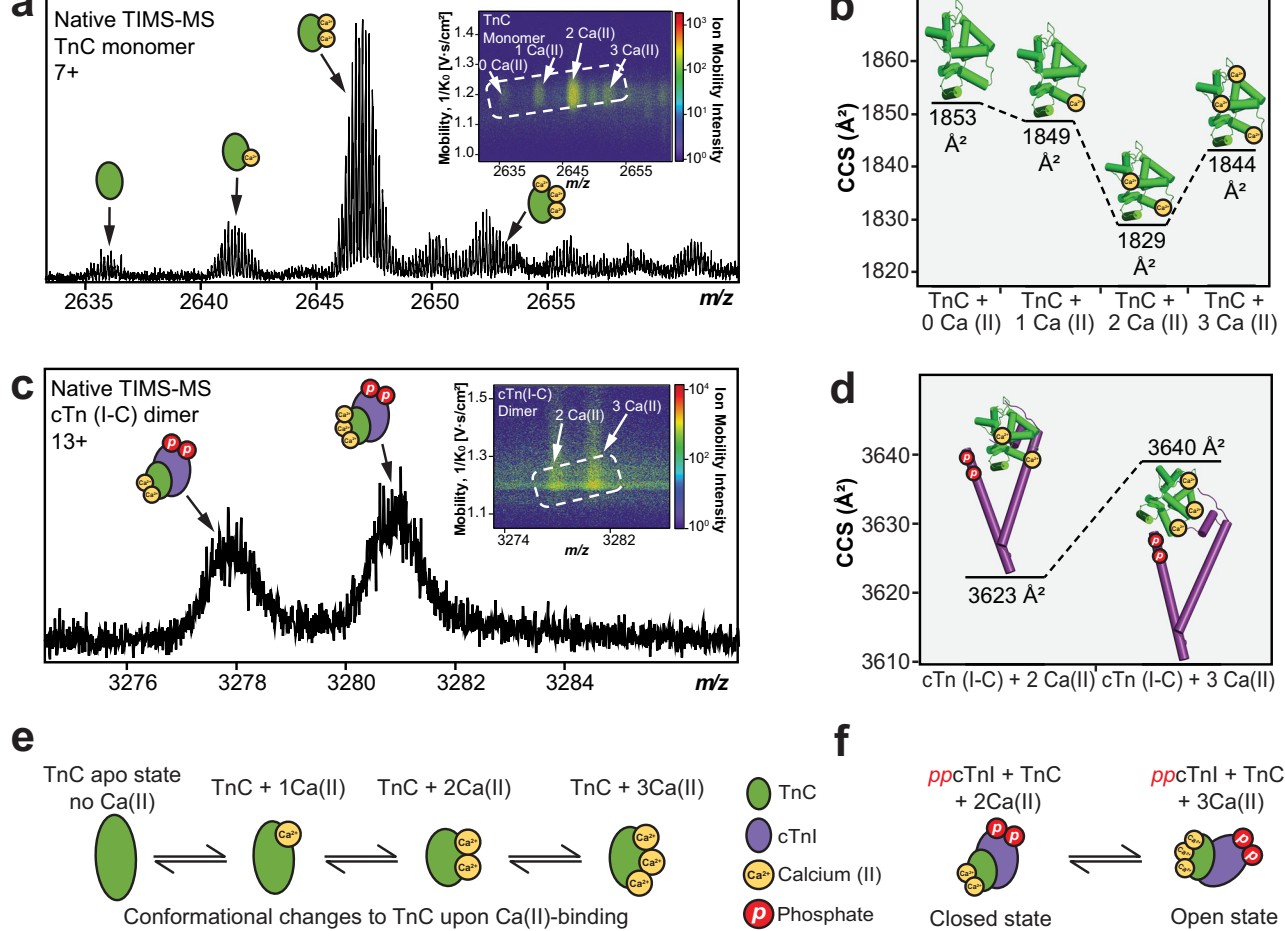

**Fig. 5 | Ion mobility separation of TnC monomer and cTn dimer Ca(II)-bound proteoforms by native TIMS-MS. a** TnC monomer proteoforms ($z = 7+$) from TIMS-MS analysis. (Inset) Associated ion mobility heat map showing differences in ion mobility and intensity for each different Ca(II)-bound proteoform. **b** Collision cross section (CCS) of the TnC proteoforms obtained under nitrogen ($N_2$) drift gas. **c** cTn dimer proteoforms ($z = 13+$) from TIMS-MS. (Inset) Associated ion mobility heat map showing differences in ion mobility and intensity for each different Ca(II)-bound proteoform. **d** CCS of the observed cTn dimer proteoforms obtained under $N_2$ drift gas. **e** Illustrations of TnC monomer and (**f**) cTn dimer conformational changes upon Ca(II)-binding.

despite incubating the cTn complex with 100 mM of EGTA, $Ca^{2+}$ may not be entirely stripped from the complex. The presence of cTnI significantly enhances the sensitivity of TnC to $Ca^{2+}$ binding, potentially preventing complete $Ca^{2+}$ removal from the cTn complex due to the equilibrium between EGTA·$Ca^{2+}$ $K_d$ and cTn·$Ca^{2+}$ $K_d$ being of a similar order. Overall, native TIMS-MS analysis of endogenous TnC monomer, cTn(I-C) dimer, and the cTn heterotrimer gas-phase structures provide insights into the conformational changes that occur upon cTn·$Ca^{2+}$ binding during cardiac contraction.

## Discussion

Here we have developed a native nanoproteomics platform for enrichment/purification and structural characterization of endogenous protein complexes and their non-covalent interactions in their native state together with comprehensive proteoform mapping. This approach addresses challenges in the current nTDMS field in the isolation and analysis of endogenous protein complexes. By avoiding denaturation or digestion steps, nTDMS quickly emerges as a powerful structural biology tool providing insights into the endogenous protein complexes and their functional states[2]. However, nTDMS studies have primarily relied on purification strategies using overexpressed recombinant proteins and/or highly abundant proteins due to the difficulties in isolating endogenous protein complexes[15–17,66–68]. To the best of our knowledge there has been no previous nTDMS study to

structurally characterize heterogenous endogenous protein complexes directly from heart tissue samples.

We chose to apply the native nanoproteomics method to characterize the structure and dynamics of the endogenous cTn complex given its high significance in cardiac function and clinical diagnosis. The structure of the cTn complex has been previously investigated by traditional structural biology techniques but were limited to resolving only the primary domains of the complex[29,34]. Moreover, it is challenging to characterize the dynamic structural changes of endogenous cTn·$Ca^{2+}$ binding events and PTMs directly from human samples by these methods[35,36,69,70]. While the influence of $Ca^{2+}$ exchange on cTn stability and function has been suggested, previous studies have often employed recombinantly expressed cTn subunits to reconstruct cTn complex or cardiac thin filaments. These constructs frequently lack PTMs and do not directly assess the entire cTn heterotrimeric complex. In contrast, our nTDMS results offer indispensable structural insights into the intact endogenous cTn complex. Notably, our native nanoproteomics study identified 17 endogenous cTn proteoforms directly from human heart tissue, while preserving PTMs including N-terminal acetylation and phosphorylation, along with non-covalent interactions such as $Ca^{2+}$ binding with high resolving power and mass accuracy. We observed sequential binding of $Ca^{2+}$ ions to TnC with the primary and secondary binding regions in domain III ([113]DLD[115]) and domain IV ([141]DKNND[145]) near the *C*-terminus, respectively. Whereas the

tertiary $Ca^{2+}$ binding region was localized to domain II ([73]DFDE[76]) at the N-terminus of TnC. $Ca^{2+}$ binding regions along TnC have been previously reported on the UniprotKb database as [65]DEDGSGTVDFDE[76] in domain II [105]DKNADGYIDLDE[116] in domain III, and [141]DKNNDGRIDY[152] in domain IV. In contrast, we provide a substantial refinement to the precise $Ca^{2+}$ binding region in TnC, to just within a few amino acid residues, which demonstrates the high-resolution capabilities of this native nanoproteomics approach for localizing non-covalent metal binding. Given that the binding of $Ca^{2+}$ to domain IV occurs in a sequential manner subsequent to the initial $Ca^{2+}$ binding event in domain III, our findings may hold considerable biological significance. Prior investigations have elucidated that mutations within the $Ca^{2+}$ binding to domains III and IV substantially diminish the affinity of TnC for the regulatory region of cTnI[71,72]. Consequently, disruption of the sequential $Ca^{2+}$ binding event could initiate deleterious physiological consequences, including the potential development of cardiomyopathies. Finally, our results resolve the structural roles of $Ca^{2+}$ binding in regulating conformational changes in TnC monomer, cTn(I-C) dimer, and cTn(I-T-C) heterotrimer.

Overall, this native nanoproteomics approach opens new opportunities for the enrichment/purification and structural characterization of endogenous protein complexes to reveal their native assemblies, proteoform landscape, and dynamical non-covalent binding. This approach utilizes functionalized NPs designed to selectively bind to specific target proteins, and the enriched protein complexes can preserve their native state when enrichment and elution conditions are established. This native nanoproteomics platform can be readily adaptable to other protein targets when the appropriate affinity ligand is identified and utilized, which will require the development and optimization of new functionalized nanomaterials. We envision the integration of designer nanomaterials with nTDMS can serve as a powerful structural biology tool in the analysis of endogenous protein complexes in their native states that are complementary to traditional biophysical techniques.

## Methods
Detailed methods are described in the *Supplementary Methods*.

### Synthesis of N-(3-(triethoxysilyl)propyl)buta-2,3-dienamide and iron-oleate precursors
N-(3-triethoxysilyl)propyl)buta-2,3-dienamide and iron-oleate were synthesized using an established method[40] with modifications. Briefly for the synthesis of -(3-triethoxysilyl)propyl)buta-2,3-dienamide, 25 mmol (2.10 g) 3-Butynoic acid, 27 mmol (7.00 g) 2-chloro-1-methylpyridinium iodide, and dichloromethane (250 mL) were added to a round bottom flask. Next, a solution of 25 mmol (5.53 g) (3-aminopropyl)triethoxysilane, 50 mmol (6.46 g) N,N-diisopropylethylamine, and dichloromethane (125 mL) was prepared separately and added to the round bottom flask. The product was concentrated in vacuo and purified twice with column chromatography to yield BAPTES and its alkyne isomer as a clear, orange oil (4.80 g, 67% yield). To effectively isomerize the propargylic isomer to N-(3-(triethoxysilyl) propyl)buta-2,3-dienamide, anhydrous acetonitrile (120 mL), 11.78 mmol (3.38 g) BAPTES isomers, and 11.78 mmol (2.50 g) tribasic potassium phosphate were added to a round bottom flask. The product was purified with flash column chromatography to yield a clear, yellow oil. (2.30 g, 68% yield). Iron oleate was synthesized by dissolving 40 mmol (10.8 g) Iron (III) chloride hexahydrate in a mixture of ethanol and nanopure water in a round bottom flask containing a magnetic stir bar. 120 mmol (36.5 g) Sodium oleate was added to the iron chloride solution along with n-hexane. Afterwards, the reaction solution was refluxed for 4-h under a $N_2$ blanket. The reaction solution was cooled, and the upper organic layer containing the iron oleate was washed three times with nanopure water. After washing, the iron oleate was concentrated *in vacuo*.

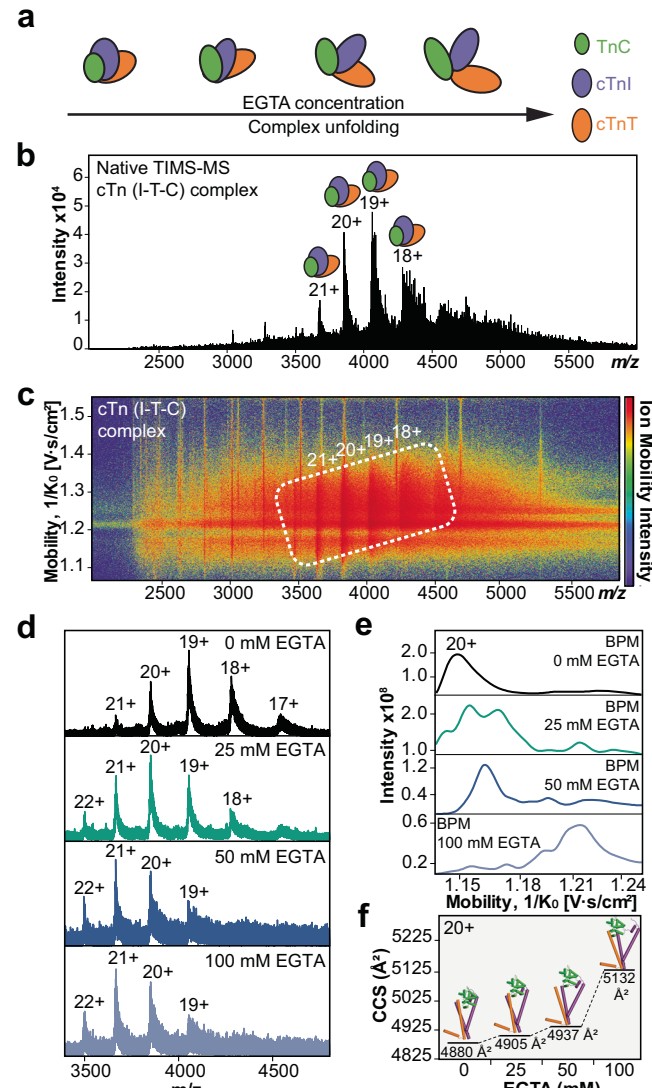

**Fig. 6 | Addition of a Ca(II) chelator to the cTn complex provides insights into cTn-Ca(II) binding dynamics. a** Illustration of the cTn complex unfolding when incubated with increasing concentrations of EGTA; a Ca(II) ion chelator that sequesters cTn-bound Ca(II). **b** Mass spectrum of the cTn complex ($z = 18$-$21+$) by native TIMS-MS. **c** Associated ion mobility heat map corresponding to the data presented in (**b**). **d** Enriched cTn complex were incubated with 0, 25, 50, and 100 mM EGTA for 3h at 4 °C and then analyzed by native TIMS-MS. **e** Base peak mobilogram spectra (BPM) of cTn complex incubated with EGTA ($z = 20+$). **f** Collision cross section (CCS) analysis of cTn complex incubated with EGTA ($z = 20+$) obtained under $N_2$ drift gas. The loss of protein complex stability directly correlates with increasing EGTA concentration (25 to 100 mM).

### Synthesis of magnetite (Fe₃O₄) nanoparticles and the functionalization by BAPTES (NP-BAPTES) and peptide (NP-Pep)
Iron oleate precursor and magnetite nanoparticles were synthesized following published protocol, with minor modifications[40]. Briefly, the NPs were synthesized using 10 mmol (9.0 g) iron oleate, 5.5 mmol (1.56 g) oleic acid, and a 4:1 ODE:TDE (40 g: 10 g) solvent mixture., the resulting NPs were then dried under vacuum and redispersed in n-hexane at a concentration of 20 mg/mL for further use. NP-BAPTES was synthesized using a previously published method[40]. $Fe_3O_4$ NPs (6 mL from a 20 mg/mL stock) were added to anhydrous n-hexane (300 mL) in a round bottom flask equipped with a Teflon-coated magnetic stir bar to achieve a total NP concentration of 0.4 mg/mL. BAPTES (1.65 mL) was added dropwise to the flask for a 0.55% (v/v)

total concentration of trialkoxysilane reagent, followed by dropwise addition of a small amount of acetic acid (30 μL) for an acidic catalyst concentration of 0.01% (v/v). After reacting at 60 °C for 24 h, the precipitate was collected and washed to remove excess silane molecules and surfactants. The NPs were then dried under vacuum for later use. 10 mg of pH-adjusted cTnI-binding peptide (HWQIAYNEHQWQC)[41] were added to 10 mg of NP-BAPTES dispersed in acetonitrile. The NP reaction mixture was allowed to react under sonication for 1 h and were then washed three times with nanopure water via centrifugation (10,000 × g, 5 min) and subsequently isolated magnetically with a DynaMag to remove unreacted peptide. The resulting peptide functionalized NPs were redispersed in nanopure water at a concentration of 5 mg/mL.

### Human cardiac tissue collection and ethical compliance
Left ventricular (LV) myocardium from non-failing donor hearts with no history of heart diseases were used and obtained from the University of Wisconsin Organ and Tissue Donation-Surgical Recovery and Preservation Services. Donor heart tissues were stored in cardioplegic solution prior to dissection and snap-frozen immediately in liquid nitrogen and stored at −80 °C. We have complied with all ethical regulations related to this study. All experiments on human samples followed all relevant guidelines and regulations. The procedures for the collection of human donor heart tissues were reviewed and approved by the University of Wisconsin-Madison Institutional Review Board (Protocol number 2013−1264). Research with human tissue has been conducted according to the principles of the Declaration of Helsinki.

### Native protein extraction
Cardiac tissue was first homogenized on ice using a Polytron homogenizer in 10 volumes (mL/g tissues) of native wash buffer (5 mM NaH$_2$PO$_4$, 5 mM Na$_2$HPO$_4$ (pH 7.0), 100 mM NaCl, 125 mM L-Met (pH 7.5), 1 mM PMSF and 1X MS-Safe protease and phosphatase inhibitor cocktail) to deplete the highly abundant cytosolic proteins. The homogenate was centrifuged at 17,000 × g for 3 min at 4 °C, and the supernatant was discarded. The washing and homogenization step was repeated once more, and then supernatant was discarded. To extract proteins such as the cardiac troponin (I-T-C) complex from human heart tissues without potentially denaturing the proteins' tertiary or quaternary structure, a high ionic strength LiCl buffer at physiological pH was used (25 mM Tris (pH 7.5), 700 mM LiCl, 125 L-Met (pH 7.5), 1 mM PMSF and 1X MS-Safe protease and phosphatase inhibitor cocktail. The resulting pellet was washed, centrifuged, and homogenized in 5 vol (mL/g tissue) of LiCl native extraction buffer, then incubated at 4 °C for 10 min to extract the sarcomeric proteome. The homogenate was centrifuged at 17,000 × g for 3 min at 4 °C and the supernatant containing the sarcomeric proteome was transferred to new Eppendorf Protein Lo-Bind tubes, and further centrifuged at 21,000 × g for 30 min at 4 °C to clarify the extracts. The supernatants were finally transferred to new Eppendorf Protein Lo-Bind tubes, snap-frozen in liquid nitrogen, and stored at −80 °C.

### Native purification of endogenous protein complexes using peptide-functionalized nanoparticle (NP-Pep) enrichment
NP-Pep (5 mg) originally dispersed in H$_2$O was redispersed in a 2 mL Eppendorf Protein Lo-Bind tube with 1 mL of equilibration buffer (25 mM Tris (pH 7.5), 700 mM LiCl, 15 mM L-Met). The NPs were then centrifuged at 21,000 × g for 2 min at 4 °C, isolated from the solution using the DynaMag, and the supernatant was removed. 1 mL of equilibration buffer was then added to the NPs, the mixture was sonicated and vortexed to prepare for protein loading. Protein loading mixture (L) from heart tissue extract was diluted to a final volume of 1 mL with a buffered solution (25 mM Tris (pH 7.5), 700 mM LiCl, 15 mM L-Met) to a total protein loading of 0.6 mg/mL and then the NP-Pep mixture was added to the protein loading mixture, at an NP concentration of

2.5 mg/ml. After this mixture was agitated on a nutating mixer at 4 °C for 1 h, the NPs were centrifuged at 21,000 × g for 2 min at 4 °C, and then isolated from the solution using the DynaMag. The supernatant was collected and saved as the flow-through (F) fraction. The isolated NPs were then washed three times with equilibration buffer (25 mM Tris (pH 7.5), 700 mM LiCl, 15 mM L-Met; 0.20 mL/mg NP-Pep) following the same centrifugation and magnetic isolation steps to remove unbound, nonspecific proteins. To elute the bound cTn(I-T-C) complex, 500 μL of 750 mM L-arginine, 750 mM imidazole, 50 mM L-glutamic acid (pH 7.5) was added, and allowed to incubate with NP-Pep with agitation on a nutating mixer at 4 °C for up to 10 min. After centrifugation and magnetic isolation, the resulting supernatant was collected as the elution fraction (E). Prior to protein concentration, Amicon Ultra Centrifugal filters were equilibrated with 500 μL of 750 mM L-arginine, 750 mM imidazole, 50 mM L-glutamic acid (pH 7.5), and centrifuged at 14,000 × g for 5 min at 4 °C. The NP-Pep elution mixture was concentrated using the Amicon Ultra Centrifugal filter (30 kDa MWCO, 0.5 mL) at 14,000 × g for 10 min at 4 °C, and the concentrated protein mixture (approximately 40 μL) was either transferred to a chilled LC-MS vial for automated online buffer exchange into 200 mM ammonium acetate solution using size exclusion chromatography columns. For offline MS analysis, concentrated protein mixture was buffer exchanged into 200 mM ammonium acetate solution using Bio-Spin® columns with Bio-Gel® P-30.

### Standard protein sample preparation
Bovine serum albumin (BSA, 66 kDa) was purchased from Sigma-Aldrich (St. Louis, MO). Native protein samples were prepared by buffer exchanging into 200 mM ammonium acetate solution by washing the sample six times through an Amicon Ultra Centrifugal filter (10 kDa MWCO, 0.5 mL). The protein sample was then diluted to 20 μM in 200 mM ammonium acetate solution.

### Calcium binding experiments using EGTA for cTn(I-T-C) complex stability
To probe the role and influence of Ca$^{2+}$ co-factor association with respect to the stability of the cTn(I-T-C) complex, a metallic ion chelator with high affinity for Ca$^{2+}$ (1.0 M EGTA, pH 7.5) was added to NP-Pep elution mixtures at various final concentrations (0, 25, 50, and 100 mM EGTA). After addition of EGTA, protein solutions were vortexed, pulse spun by tabletop centrifuge, and incubated on ice at 4 °C for 3 h. After incubation with EGTA, samples were concentrated, and buffer exchanged into 200 mM ammonium acetate solution for offline native MS analysis as described above.

### Top-down RPLC-MS/MS analysis of cTnI, cTnT, and TnC monomers
Top-down RPLC-MS/MS was carried out by either using an Acquity ultra-high pressure LC M-class system (Waters) coupled to a high-resolution maXis II quadrupole time-of-flight (QTOF) mass spectrometer (Bruker Daltonics) or by using a nanoAcquity ultra-high pressure LC system (Waters) coupled to a high-resolution Impact II QTOF mass spectrometer (Bruker Daltonics). 600 ng of total protein (n = 3) was injected onto a home-packed PLRP column (PLRP-S) (Agilent Technologies), 10-μm particle size, 500-μm inner diameter, 1,000 Å pore size using an organic gradient of 20 to 65% mobile phase B (mobile phase A: 0.2% FA in H$_2$O; mobile phase B: 0.2% FA in 50:50 acetonitrile/isopropanol) at a constant flow rate of 12 μL/min.

### Size exclusion chromatography (SEC) for online buffer exchange (OBE) and native MS analysis of the cTn(I-T-C) complex
SEC experiments were performed using a NanoAcquity ultra-high pressure LC system (Waters) coupled to a high-resolution maXis II quadrupole time-of-flight mass spectrometer (Bruker Daltonics). 1 μg

of total protein (n = 3) was injected onto a PolyHYDROXYETHYL A column (PolyHEA) (PolyLC Inc), 2.1 mm internal diameter, 100 mm length, 5 μm particle size, and 200 Å pore size. Protein samples were separated isocratically with 200 mM ammonium acetate solution at a flow rate of 28 μL/min for 10 min with a 'divert to waste' step programmed at 7 min.

### nTDMS offline FTICR analysis of cTn complex

Samples were analyzed by nanoelectrospray ionization via direct infusion using a TriVersa Nanomate system (Advion BioSciences) coupled to a solariX XR 12-T Fourier Transform Ion Cyclotron Resonance mass spectrometer (FTICR-MS, Bruker Daltonics). For the nanoelectrospray ionization source using a TriVersa Nanomate, the desolvating gas pressure was set to 0.60 PSI and the voltage was set to 1.5–1.7 kV versus the inlet of the mass spectrometer. The source dry gas flow rate was set to 4 L/min at 180°C. For the source optics, the capillary exit, deflector plate, funnel 1, skimmer voltage, funnel RF amplitude, octupole frequency, octupole RF amplitude, collision cell RF frequency, and collision cell RF amplitude were optimized at 190 V, 200 V, 150 V, 120 V, 300 Vpp, 2 MHz, 600 Vpp, 1.4 MHz, and 2000 Vpp, respectively. Mass spectra were acquired with an acquisition size of 1 to 4M-words of data in the mass range 200-8000 m/z. Ions were accumulated in the collision cell for 5 to 30 s, and a time-of-flight of 2 ms was used prior to their transfer to the ICR cell. For collisionally activated dissociation (CAD) tandem MS (MS/MS) experiments, the collision energy was varied from 55 to 70 V, ion accumulation was optimized to 5 to 50 s, and acquisition size varied from 1 to 4M-words of data.

### Native trapped ion mobility (TIMS) MS analysis of cTn complex

Samples were directly infused by a syringe at a flow rate of ~3 μL/min into a timsTOF Pro mass spectrometer (Bruker Daltonics). For the MS inlet, the end plate offset and capillary voltage were set to 500 V and 3800 V, respectively. The nebulizer gas pressure ($N_2$) was set to 1.5 bar with a dry gas flow rate of 6.0 L/min at 180°C. The tunnel out, tunnel in, and TOF vacuum pressures were set to 8.46E-01, 2.586, and 1.52E-07 mBar. TIMS funnel 1 RF was set to 350 Vpp, and TIMS collision cell energy was set to 125 V. An IMS imeX accumulation time of 5.0 ms and cycle ramp time of 350 ms were found to yield optimal resolving power. The TIMS accumulation time was locked to the mobility range (typically 1.05–1.55 $1/K_0$). In the MS transfer optics, the funnel 1 RF, funnel 2 RF, deflection delta, isCID energy, multipole RF, and quadrupole ion energy were optimized to 350 Vpp, 350 Vpp, 60 V, 80 eV, 550 Vpp, and 2 eV. Agilent tune mix was directly infused to calibrate MS and CCS values. For MS calibration, the MS resolution for the most abundant calibrant signal, 1821 m/z, was 58, 000. Calibrant points at 922, 1222, and 1522 m/z were used for TIMS calibration. The TIMS resolution for the most abundant calibrant signal, 1821 m/z, was 70.6 CCS/ΔCCS. For native MS spectral collection, the quadrupole low mass was set to 2000 m/z with a scan range of 2000-8000 m/z. The collision energy was set to 4 eV, with a 2000 Vpp collision cell RF, a 232 μs transfer time, and a prepulse storage time of 5.0 μs.

### Data analysis

Mass spectrometry data was collected using otofControl v. 4.3 and ftmsControl v. 2.1.0. All data were processed and analyzed using Compass DataAnalysis v. 4.3/5.3 software and MASH Native v. 1.1[73]. The sophisticated numerical annotation procedure (SNAP) peak-picking algorithm (quality factor 0.4; signal-to-noise ratio (S/N) 3.0; intensity threshold 500) was applied to determine monoisotopic mass of all detected ions. All chromatograms were smoothed by the Gauss algorithm with a smoothing width of 2.04s. Denatured mass spectra were deconvoluted using the Maximum Entropy algorithm within the DataAnalysis v. 4.3 software with the resolving power for deconvolution set to 40,000 or 60,000k. MS/MS data were output from the DataAnalysis software and analyzed using Native MASH for

proteoform identification, calcium localization, and sequence mapping. All fragment ions were manually validated with a mass tolerance of 20 ppm. For experimental ion mobility analysis by timsTOF Pro, the collisional cross section (CCS) in A² for a species of interest was determined via the Mason Schamps equation (Eq. 1).

$$\text{CCS} = \frac{3}{16}\sqrt{\frac{2\pi}{\mu k_b T}}\frac{ze}{N_0 k_0} \tag{1}$$

where μ is the reduced mass of the ion−gas pair ($\mu = \frac{mM}{(m+M)}$, where m and M are the ion and gas particle masses), $k_b$ is Boltzmann's constant, T is the drift region temperature, z is the ionic charge, e is the charge of an electron, $N_O$ is the buffer gas density, and $k_O$ is the reduced mobility. Theoretical CCS values were determined using the IMPACT method[46].

### Statistical analysis

Statistical analysis for group comparison was completed using paired Student's t tests. All p-values at $p < 0.01$ were considered significant. All error bars indicated in the figures represent the mean ± standard error of the mean (SEM).

### Reporting summary

Further information on research design is available in the Nature Portfolio Reporting Summary linked to this article.

## Data availability

The mass spectrometry proteomics data generated in this study have been deposited to the ProteomeXchange Consortium via the PRIDE partner repository under the accession code PXD042825 and MassIVE repository under the accession code MSV000092130. In addition, all the raw data files or spectra are available upon request. The structures corresponding to the PBD codes mentioned in the main text are available through these links: Figs. 1 and 4: 1J1E. Source data are provided with this paper.

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

## Acknowledgements

This work was supported by the National Institute of Health (NIH) R01 GM117058 (to both Y.G. and S.J.). Y.G. also would like to acknowledge R01 HL109810, R01 GM125085, and S10 OD018475. E.A.C. and T.N.T. would like to acknowledge support from the NIH Chemistry-Biology Interface Training Program NIH T32GM008505. D.S.R. would like to acknowledge support from the American Heart Association Predoctoral Fellowship Grant No. 832615/David S. Roberts/ 2021. J.A.M. would like to acknowledge support from the Training Program in Translational Cardiovascular Science T32HL007936. We would like to thank Boris Krichel for the helpful discussions during revision of the manuscript. We thank James Anderson and Carrie Sparks at the University of Wisconsin Organ and Tissue Donation- Surgical Recovery and Preservation Services for coordination of donor heart collection.

## Author contributions

E.A.C., D.S.R., T.N.T., S.J., and Y.G. designed research; E.A.C., D.S.R., and T.N.T. performed MS analysis; E.A.C., T.N.T., and B.H.L. prepared the biological samples; D.S.R., T.N.T., J.A., M.W., E.A.R., D.K. synthesized and characterized functional nanoparticles; A.J.A. contributed analytic tools; E.A.C., D.S.R,. T.N.T., J.A., M.W., E.A.R., S.J., and Y.G. analyzed data; and E.A.C., D.S.R., T.N.T., S.J., and Y.G. wrote the paper with comments from other co-authors.

## Competing interests

The University of Wisconsin-Madison has filed a provisional patent application serial No.62/949,869 (December 18, 2019) and US Patent App. 17/786,482, (Feb. 2, 2023) based on this work. Y.G., S.J., D.S.R., and T.N.T., are names as the inventors on the provisional patent application. The late Dr. Andrew J. Alpert was the former President and founder of PolyLC, Inc. The other authors declare no competing interests.
