## [Peer Review File · Nature Communications]

Structure and dynamics of endogenous cardiac troponin complexes in human heart tissue captured by native nanoproteomicsREVIEWER COMMENTS

Reviewer #1 (Remarks to the Author):

The manuscript describes the development of a "native nanoproteomics" strategy to study low-abundance protein complexes directly from tissues using surface-functionalized superparamagnetic nanoparticles and native top-down mass spectrometry (nTDMS). The researchers applied this method to comprehensively characterize the structure and dynamics of the cardiac troponin (cTn) complex directly from human heart tissues, providing insights into its stoichiometry, conformational changes, and proteoform landscape, which has not been effectively captured by conventional structural biology techniques.

The strength of this method lies in its elegant purification approach, which utilizes a specific interaction between a peptide and the cTnI subunit of the cTn complex (previously described in Nat. Comm. 2020). In the current study, the authors successfully employed this specific interaction to isolate the entire cTn complex under native conditions and analyze it using native mass spectrometry, yielding valuable insights. However, the authors' claim of presenting a generic analysis method for "Structure and dynamics of endogenous protein complexes" in the title, abstract, and introduction is not fully supported, as this study focuses solely on the cTn complex. Given that the use of beads and the combination of native and top-down MS are not novel, I feel it would be more appropriate for this study to be published in a more specialized journal.

Furthermore, the authors claim that their developed method is suitable for "low abundance protein complexes." However, in the example they present of the cTn complex, the individual subunits (Tnc, Tnl, and cTnT) appear to be among the most abundant proteins in the lysate (Fig. 2a). Additionally, supplementary Figure S3 demonstrates that RP LC/MS of the lysate itself identified the three cTn proteins without any purification. This raises the question of whether the method is truly relevant for low-abundance proteins.

Additional comments:

- It would have been more appropriate for the authors to acknowledge the origin of the peptide in this paper, as it was previously developed and characterized by the Naik group (ACS Sens. 2018). Later on the group employed it in the previous publication (Nat. Comm. 2020).
- Regarding the MS/MS experiments presented in Figures 3 and 4, the authors should address whether they can detect any differences in the ejection of modified subunits compared to non-modified ones. Does calcium binding and/or phosphorylation have any impact on the stability of the protein complex.
- In the MS/MS experiments (Figures 3, 4 and Supplementary Figure 8), it is unclear why the stripped complexes are not shown?
- Please comment on why cTnI is not stripped from the complex during MS/MS analysis?
- Given that cTnI is not detected, nor do the stripped complexes, the validity of the calculation in Supplementary Figure 8b is not clear.
- Figure 3, the dashed lines that connects the insets to the figure is not clear and confusing.
- On page 13, the authors discuss "collisional activation ramping" experiments aimed at identifying the "least" and "most" Ca²⁺ binding region. However, the basis for their conclusion regarding the order of sequential binding of calcium ions remains unclear. It is important to understand why they assume that

sensitivity to collision activation correlates with the binding order of calcium ions.

- From the top-down data how were the 3-5 amino acids domains of calcium binding defined, given that the fragmentation map indicates longer peptides?
 - The data of the “collisional activation ramping” experiments is not shown. Only a single spectrum.
 - On page 17 it is indicated that “17 endogenous cTn proteoforms” were identified, however, only 10 cTn(I-T-C) proteoforms are listed in Supplementary table 2.
 - In supplementary Figure 2, TnC seems to be proteolytically degraded (batch 2 and 3), can the authors comment on this?
 - How was the relative abundance in Supplementary Figure 4 calculated?
 - The entire scheme of OBE and the inclusion of panels b-d in Supplementary Figure 5 is unclear, as these results have already been published by the lab of Vicki Wysocki.
 - Figure 5, panels c and d - what do the dashed lines represent? What m/z range was isolated for the TIMS-MS experiments. Do they cover only one proteoform? How was the C-terminal lysine deletion in cTnT validated? What specie of the dimer was isolated for the validation of the double phosphorylation?
 - In the ion mobility experiments in figure 5, how can the conditions for BSA analysis, a monomer without cofactors, assist in optimizing conditions for analysis of the complex, bound to Calcium?
- In figure 5, panels e and f are not required, they simply describe panels b and d in a different type of representation. Same goes for panel a in figure 6, which recapitulates panel e, and the structure of EGTA is completely irrelevant here. Also – what are the measured masses of the complexes in panel d? Can the authors deduce information on the level of calcium binding from the measured masses?

Reviewer #2 (Remarks to the Author, also attached as PDF):

Chapman and coauthors developed a “native nanoproteomics” strategy for the enrichment of native cardiac troponin (cTn) complexes directly from human heart tissue using peptide-functionalized superparamagnetic nanoparticles under non-denaturing conditions, native top-down mass spectrometry (nTDMS) characterization of low abundance cTn complexes was subsequently performed, which yields the stoichiometry and composition of the heterotrimeric cTn complex, localizes Ca²⁺ binding domains (II-IV), defines cTn-Ca²⁺ binding dynamics, and provides high-resolution mapping of the proteoform landscape.

Overall, this is an exciting work that opens a new paradigm for structural characterization of low-abundance native protein complexes. This manuscript is well written and the experimental is well designed. There are a few major concerns going back to the results as outlined below, along with some minor issues listed below that need to be addressed prior to publication:

Major:

1. When we look at Figure 2b, 2c-d, and Figure 3d together, there is something strange. As shown in Figure 2c and 2d, the most abundant cTn(I-T-C) complexes are cTnT(p)-cTnI(p)-TnC(2Ca²⁺), cTnT(p)-cTnI(p)-TnC(3Ca²⁺), cTnT(p)-cTnI(2p)-TnC(2Ca²⁺), and cTnT(p)-cTnI(2p)-TnC(3Ca²⁺); while Figure 3d

displays that the ejected cTn(I-C) dimers are cTnI-TnC(2Ca²⁺), cTnI-TnC(3Ca²⁺), cTnI(p)-TnC(2Ca²⁺), cTnI(p)-TnC(3Ca²⁺), cTnI(2p)-TnC(2Ca²⁺), and cTnI(2p)-TnC(3Ca²⁺), among which, the ejected cTnI-TnC(2Ca²⁺) and cTnI-TnC(3Ca²⁺) dimers (the dimers with the wide type cTnI) are the most intensive ones, but their corresponding cTn(I-T-C) complexes (cTnT(p)-cTnI-TnC(2Ca²⁺) and cTnT(p)-cTnI-TnC(3Ca²⁺)) are of low abundance in Figure 2d (not labelled). Please address how this pheromone happened. Likewise, the abundance ratio for monomeric cTnI: cTnI(p) : cTnI(2p) is ~ 1:1:0.5 (Figure 2b); while when we sum the abundances of cTnI-TnC(2Ca²⁺) and cTnI-TnC(3Ca²⁺) for cTnI, cTnI(p)-TnC(2Ca²⁺) and cTnI(p)-TnC(3Ca²⁺) for cTnI(p), and cTnI(2p)-TnC(2Ca²⁺) + cTnI(2p)-TnC(3Ca²⁺) for cTnI(2p), respectively, the abundant ratio for cTnI: cTnI(p) : cTnI(2p) is 1:0.5:0.15 (Figure 3d); the abundant ratio for cTnI(p) : cTnI(2p) in Figure 2d is about 1:1 for cTnT(p)-cTnI(p)-TnC(2Ca²⁺) + cTnT(p)-cTnI(p)-TnC(3Ca²⁺), cTnT(p)-cTnI(2p)-TnC(2Ca²⁺) + cTnT(p)-cTnI(2p)-TnC(3Ca²⁺). please explain why would the abundance ratios of cTnI: cTnI(p) : cTnI(2p) at monomeric forms, cTn(I-C) dimers, and cTn(I-T-C) complexes vary so differently?

2. Page 13, in the following discussion, “Progressive collisional activation ramping revealed TnC domain III to be the least vulnerable Ca²⁺ binding to increasing collisional activation, while domain II was found to be the most vulnerable Ca²⁺ binding region. To localize the primary region for Ca²⁺ binding that is least vulnerable to collisional activation, we isolated the TnC proteoform at 2312 m/z and performed CAD to yield product ions y₅₂ + Ca²⁺, y₃₀, b₁₁₅ + Ca²⁺, and b₁₀₉ (Figure S12). Therefore, the primary Ca²⁺ binding domain was localized to 113DLD115 in domain III. The next Ca²⁺ binding domain was localized to the structural region between 141DKNND145 in domain IV by first isolating the TnC proteoform at 2316 m/z and then generating CAD product ions b₁₄₀, b₁₄₅ + Ca²⁺, y₁₆, and y₂₂ + Ca²⁺ (Figure S13). Finally, the most vulnerable Ca²⁺ binding region was localized to regulatory domain II between 73DFDE76 by isolating the TnC proteoform at 2321 m/z and generating CAD product ions b₆₅, b₉₁ + Ca²⁺, y₈₅, and y₉₄ + Ca²⁺ (Figure S14).”, it seems that the authors have some prior knowledge about the binding order of each Ca²⁺ ion, if yes, please clarify and add in corresponding references. Otherwise, it is rather strange to conclude that the binding strength can be related to specific domain through progressive collisional activation ramping. Additionally, the authors performed CAD experiments for TnC+1Ca at m/z 2312 (Figure S12), TnC +2Ca at m/z 2312 (Figure S13), and TnC +3Ca at m/z 2312 (Figure S14), respectively, and observed the primary Ca²⁺ binding domain as domain III. Did the authors see any Ca²⁺ binding fragments related to domain IV in Figure S12? If not, does this mean that the Ca²⁺ binding to domain IV is a sequential event upon the primary Ca²⁺ binding to domain III? Any biological significance? If the Ca²⁺ ion was found to bind to domain IV in CAD of TnC+1Ca, would that an indication of malfunction of TnC? Furthermore, for the CAD results for TnC+2Ca(II), it states “The next Ca²⁺ binding domain was localized to ... 141DKNND145 in domain IV ... generating CAD product ions b₁₄₀, b₁₄₅ + Ca²⁺, y₁₆, and y₂₂ + Ca²⁺ (Figure S13)”, the N-terminal product ions by AA140 should cover the primary Ca²⁺ binding site at domain III and AA145 should bind to the two Ca²⁺ ions bound to domain III and IV, something like, b₁₄₀ + Ca²⁺ and b₁₄₅ + 2Ca²⁺. Similarly, “Finally, the most vulnerable Ca²⁺ binding region was localized to regulatory domain II between 73DFDE76 by isolating the TnC proteoform at 2321 m/z and generating CAD product ions b₆₅, b₉₁ + Ca²⁺, y₈₅, and y₉₄ + Ca²⁺,” y₈₅ and y₉₄ should cover 2Ca²⁺ and 3 Ca²⁺ binding sites, respectively. Something like y₈₅+ 2Ca²⁺, and y₉₄+ 3Ca²⁺. Please replace the insert figures of these fragment ions with the ones that can simultaneously reflex other binding sites. Last but not least, there are mistakes in figure legends for Figure S12, S13, and S14. For example, it states “Representative collisionally activated dissociation (CAD) fragment ions (y₃₀ 2+, y₅₂ 2+, b₁₀₉ 6+, b₁₁₅ 6+)” in the manuscript, but in Figure S12(b), it says “Representative collisionally activated dissociation

(CAD) fragment ions (y_{30}^{2+} , y_{52}^{2+} , b_{109}^{6+} , b_{115}^{6+}) obtained from the nTDMS analysis.". The same mistakes also present in S13 and S14.

3. As shown in Table S1, the authors obtained and studied a list of 5 non-failing donor hearts from clinical donors with different causes of death, but in the manuscript, whether or not their disease conditions and causes of death affect the existing forms, stability, and dynamics of cTn complexes were not mentioned. It would be good to add in some discussion for this part.

Minor:

4. Page 4, "However, only partial structural information has been obtained from conventional X-ray crystallography excluding the intrinsically disordered but functionally critical regions of cTnI and cTnT. Moreover, the cTn structure is highly dynamic due to Ca^{2+} binding^{23, 27, 28} and PTMs^{9, 29, 30} that regulates muscle contraction, yet traditional methods have not effectively captured these dynamic conformational changes³¹. Furthermore, recombinantly expressed proteins have been used in previous studies thus important structural features vital to the function of the endogenous cTn complex within the sarcomere were lost^{32, 33}" The introduction about the structure information of cTn complex is a bit vague, it is better to specify what kind of information has been obtained by biophysical approaches such as X-ray, cryoEM, and what has been lost. It will in turn strength the significance of this work.

5. page 9, in Figure 2d, for the zoomed-in mass spectra from c is within m/z 4057 – 4067; therefore, the dash line displays the zoomed-in region should be corrected accordingly.

6. page 10, "In-depth examination of the endogenous cTn complex revealed four unique proteoforms comprised of both covalent and non-covalent modifications (Figure 2d)" should be "... four unique proteoform complexes ...".

7. page 10, "mono- and bis-phosphorylated cTnI, and TnC with three bound Ca^{2+} ions (most abundant cTn complex MW = 77136 Da)" should be "mono-phosphorylated cTnI".

8. Page 11, "Our nTDMS analysis also suggests that both the intrinsically disordered C- and N-termini of cTnI are more solvent exposed than the stable internal regions that form the subunit-subunit interfaces of the cTn complex.", corresponding reference(s) should be added.

9. Page 15, "Due to the conformational heterogeneity and presence of intrinsically disordered regions along the heterotrimeric cTn complex, it is challenging to obtain crystal structures of cTn in its active and closed states upon Ca^{2+} binding using traditional structural biology techniques⁴³. Please specify what does the "closed states" exactly mean?"

10. Page 15, "The CCS values for TnC monomer ... for the most abundant charge state ($z = 8+$), respectively (Figure 5a-b)."; it should be $z=7+$. Similarly, "On the other hand, the CCS values for cTn(I-C) dimer with 2 (3623 Å²) and 3 (3640 Å²) Ca^{2+} ions for the most abundant charge state ($z = 15+$) ... (Figure 5c-d, Table S3)", it should be $z=13+$.

11. Page 17, "conformation to prepare the N-terminal region of TnC for binding to the C-terminal region of cTnI to initiate cardiac muscle contraction. Corresponding reference(s) should be added.

Reviewer #3 (Remarks to the Author):

Dear author,

Thank you for the comprehensive description of the antibody-free top-down approach to specifically enrich the low-abundance troponin complex. The manuscript is very well written; clear, precise, and easy to understand. Some text is repetitive from the previous publication <https://doi.org/10.1038/s41467-020-17643-1> but I understand that the point has to be made. The following minor suggestions are

Minor suggestions:

1. There method has been developed and described by the group before, please rewrite the sentence at page 4:

“Here, we have developed a “native nanoproteomics” platform integrating the native enrichment of low-abundance protein complexes directly from tissues using surface functionalized superparamagnetic nanoparticles (NPs) with high-resolution nTDMS to characterize the structure and dynamics of low-abundance endogenous protein complexes for the first time”.

Author used different name to describe the top-down method, current manuscript: native top-down mass spectrometry (nTDMS) compared to previous manuscript: top-down LC/MS coupling reversed-phase liquid chromatography (RPLC) to high-resolution MS. However, based on the method section both manuscripts used the same instruments and set up to run the samples for the general cTn identification. Therefore, I encourage author to acknowledge previous publication. I do understand that previous manuscript concentrated on TnI however as described and presented, for example in supplement figure 4 and 7 the whole cTn complex was enriched and the peptide used to functionalize the NP surface was the same for both experiments/manuscripts.

2. Did author measure the cTn percent recovery of the NP-Pep?

3. Grammar: page 3. “These present tremendous challenges to studying their structure and dynamics using...” should be “to study”

4. Please rewrite the sentence, it does not make sense, page 4: “Furthermore, recombinantly expressed proteins have been used in previous studies thus important structural features vital to the function of the endogenous cTn complex within the sarcomere were lost”.

5. Please rewrite the sentence, as verb is missing; figure 4 : ” (d) Structural representation of the cTn complex with the three Ca(II)-binding domains (II-IV) discovered in these experiments highlighted (domain II, blue; domain III, red; domain IV, yellow, UniprotKB annotations, gray).

6. Please avoid text repetition, e.g. introduction creeping into the discussion part: “investigated by X-ray, crystallography, NMR, and cryo-EM”, page 3, 4, and 19. The problem was described in the introduction, it does not have to be repeated one more time.

7. In the discussion maybe author can compare the phosphorylation analysis between current NP results to results from affinity purification performed by the same group (Tiambeng TN, Tucholski T, Wu Z, Zhu Y, Mitchell SD, Roberts DS, Jin Y, Ge Y. Analysis of cardiac troponin proteoforms by top-down mass spectrometry. *Methods Enzymol.* 2019;626:347-374. doi:10.1016/bs.mie.2019.07.029).

8. Author use PTMs but described only phosphorylation. Please keep in mind other modifications, including acetylation, methylation, oxidation, among others were reported on the troponin's.

Responses to Reviewer Comments

Reviewer #1:

The manuscript describes the development of a "native nanoproteomics" strategy to study low-abundance protein complexes directly from tissues using surface-functionalized superparamagnetic nanoparticles and native top-down mass spectrometry (nTDMS). The researchers applied this method to comprehensively characterize the structure and dynamics of the cardiac troponin (cTn) complex directly from human heart tissues, providing insights into its stoichiometry, conformational changes, and proteoform landscape, which has not been effectively captured by conventional structural biology techniques.

The strength of this method lies in its elegant purification approach, which utilizes a specific interaction between a peptide and the cTnI subunit of the cTn complex (previously described in Nat. Comm. 2020). In the current study, the authors successfully employed this specific interaction to isolate the entire cTn complex under native conditions and analyze it using native mass spectrometry, yielding valuable insights. However, the authors' claim of presenting a generic analysis method for "Structure and dynamics of endogenous protein complexes" in the title, abstract, and introduction is not fully supported, as this study focuses solely on the cTn complex. Given that the use of beads and the combination of native and top-down MS are not novel, I feel it would be more appropriate for this study to be published in a more specialized journal.

Response: We thank the Reviewer for the overall positive evaluation of our work and the constructive comments. Indeed, this native nanoproteomics approach combining antibody-free functionalized nanoparticles for native protein complex enrichment and native top-down mass spectrometry can be generally applied. The generality comes from the general approach of functionalized nanoparticles with specific binding to targeted protein(s) and the demonstration herein that the enriched protein complex can maintain native state when suitable enrichment and elution conditions are developed. The specifically demonstrated nanoparticles herein are designed for enriching cTn complexes. We are indeed applying this approach to more protein systems such as kinases (e.g. AMPK protein complex) and receptors (e.g. ACE2 receptors). We have some preliminary proof-of-principle data and plan to publish such additional applications of this native nanoproteomics method in the future when they are fully executed. But each system will be an independent paper with new nanomaterials development, in a similar fashion as other native mass spectrometry studies for structural biology applications.¹⁻⁵

Moreover, we would like to clarify the nanoproteomics method we *previously described in Nature. Comm. 2020* only applied to **denatured** top-down proteomics with a focus on the enrichment of low-abundance proteoforms in human serum samples for **clinical applications**. In contrast, here we develop a novel **native** nanoproteomics method to enrich native protein complexes directly from tissues for **structural biology applications** which has a completely different focus from the previous paper. Notably, this work represents the *first study* to comprehensively characterize the structure and dynamics of *endogenous* protein complexes directly from human tissues using nTDMS that led to new structural biology information and biophysical insights. And this is accomplished by carefully integrating rationally designed nanomaterials with native top-down

MS. We believe that this highly interdisciplinary study is of great interest to the diverse readership of *Nature Communications*, including but not limited to the structural biology, protein biochemistry, mass spectrometry, native proteomics, and bio-nanotechnology research communities.

We have revised the manuscript to include more discussion and explanations to clarify these points (Page 21-22) and toned down the claim on general applicability by removing “general” on Page 6.

Furthermore, the authors claim that their developed method is suitable for "low abundance protein complexes." However, in the example they present of the cTn complex, the individual subunits (Tnc, TnI, and cTnT) appear to be among the most abundant proteins in the lysate (Fig. 2a). Additionally, supplementary Figure S3 demonstrates that RP LC/MS of the lysate itself identified the three cTn proteins without any purification. This raises the question of whether the method is truly relevant for low-abundance proteins.

Response: We thank the Reviewer for this constructive and thoughtful comment. First, we would like to clarify that the lysate shown in the original Figure 2a (now Figure S3a) is already the result of multiple depletions of highly abundant proteins. For our native protein extractions, we must first deplete highly abundant cytosolic proteins using multiple rounds of native phosphate wash buffer. We have made further clarifications in the Methods section (Page 26). Following cytosolic protein depletion, we further enrich sarcomeric proteins using a high ionic strength LiCl extraction buffer at physiological pH. Finally, from the enriched sarcomeric proteins, we perform specific enrichment of cTn using NP-Pep to isolate the intact cTn(I-T-C) complex from the rest of the sarcomeric proteins. We performed the *denatured* SDS-gel (Figure S3a) and RPLC-MS (now Figure S4) experiment to essentially serve as a quality control of the NP-Pep before performing our native top-down MS studies. While we do observe some *denatured* TnC, cTnI, and cTnT subunits in Figure S3 and Figure S4, they are not the most abundant proteins present in the LiCl loading mixture (LM). The point of Figure S4 was to demonstrate that all cTn subunits are significantly enriched in the elution mixture (EM) compared to the LM.

More importantly, we would like to clarify that, *without the use of NP-Pep, we were not able to enrich and characterize the native cTn heterotrimer complex directly from human cardiac tissue.* We have performed an experiment directly infusing native sarcomeric LM into the FT-ICR to see if we could observe native cTn complex directly in the LM without any purification. Our results in Figure R1 (shown below; for Reviewer only) show that the cTn(I-T-C) complex is ***not observed*** even when sarcomere-enriched LiCl lysate is directly infused for FT-ICR MS analysis. Instead, the more highly abundant sarcomeric proteins such as myosin light chain (MLC) 2v, MLC 1v, and tropomyosin are mainly observed. Therefore, these experiments make it clear that the cTn(I-T-C) complex is indeed present in low abundance in our lysate, and enrichment/purification strategies such as NP-Pep are indeed needed to comprehensively characterize the complex by native top-down MS.

Nevertheless, we have toned down the claim on the low-abundance proteins in various places throughout the revised manuscript by replacing the word of “low-abundance” with “endogenous”.

Figure R1: Native mass spectrum of sarcomere-enriched LiCl loading mixture (LM) using FTICR-MS. Desalted native LiCl LM in 150 mM Ammonium Acetate was analyzed by nanoelectrospray ionization via direct infusion using a TriVersa Nanomate system coupled to a solariX XR 12-T FTICR-MS. MLC-2v (black), MLC-1v (red), and tropomyosin (purple) were the most abundant sarcomeric proteins observed in the LM, and cTn complexes were not detected between ~ 3800 - 4800 m/z .

1. *It would have been more appropriate for the authors to acknowledge the origin of the peptide in this paper, as it was previously developed and characterized by the Naik group (ACS Sens. 2018). Later on the group employed it in the previous publication (Nat. Comm. 2020).*

Response: We thank the Reviewer for the thoughtful suggestion. We have now included the appropriate citations in the main text, methods section, and Supplementary Information to acknowledge the origin of the peptide design in our work and the exemplary work performed by the Naik group to develop and characterize this peptide.

2. *Regarding the MS/MS experiments presented in Figures 3 and 4, the authors should address whether they can detect any differences in the ejection of modified subunits compared to non-modified ones. Does calcium binding and/or phosphorylation have any impact on the stability of the protein complex.*

Response: We thank the Reviewer for the excellent comments. We do notice some differences in the ejection of modified versus non-modified cTn subunits that have allowed us to infer the relative stability of the different cTn molecular forms based on how they responded to the nTDMS analysis. In particular, the cTn(I-C) dimer which consisted of primarily bis-phosphorylated cTnI + TnC with two and three Ca^{2+} ions bound was extremely robust, needing high collision voltage (70 V) to generate sequence informative fragment ions during the complex-down analysis. On the other hand, only 55 V of collision voltage was needed to generate sequence informative fragment ions of TnC in all its Ca^{2+} bound states during complex-down analysis. Finally, only 12 V of collision

voltage was needed to eject cTn subunits from the cTn(I-T-C) complex during complex-up analysis. From these data we can infer that cTn(I-C) dimer is highly resilient to collisional activation and is extremely stable, whereas the cTn(I-T-C) complex is less resilient and stable. It is extremely challenging to isolate each specific Ca²⁺ bound/phosphorylated proteoform for the cTn(I-C) heterodimer and the cTn(I-T-C) heterotrimer in our FTICR so we could not infer exactly how Ca²⁺ binding and phosphorylation impacted the stability of the protein complexes.

3. In the MS/MS experiments (Figures 3, 4 and Supplementary Figure 8), it is unclear why the stripped complexes are not shown?

Response: We thank the Reviewer for the constructive comment. In Figure 3b, the complex is shown between ~3800-4300 m/z during the complex-up native MS experiment which aims to provide information about complex stoichiometry without backbone cleavage.⁶ During our complex-up analysis, unfolded and highly charged cTn subunits were ejected from the complex during broadband CAD. Typically, noncovalent complexes subjected to CAD result in the asymmetric dissociation of oligomers into highly charged ions at lower m/z and lowly charged ions at higher m/z (e.g. charge stripped complexes).⁷ We did not observe these lowly charged “stripped” cTn heterotrimer complexes which is expected at a much higher m/z likely due to the limitation of our 12 T Bruker FTICR mass spectrometer which is not optimized for detection of ions at very high m/z .⁸ We only observed a substantial decrease in intensity of the cTn complex when activated by CAD (Figure 3b) as opposed to Figure 3a when no CAD activation occurred.

4. Please comment on why cTnI is not stripped from the complex during MS/MS analysis?

Response: We thank the Reviewer for the insightful comment. Previous studies have found that the binding of cTnI to the C-terminal domain of TnC is very robust with nanomolar dissociation constant.^{9, 10} Moreover, circulating cTnI is found in human serum existing primarily as a heterodimer with TnC.¹¹ In all of our complex-up experiments, we did not detect ejected cTnI monomer in appreciable abundance, as we saw mostly cTn(I-C) dimer. We suspect that this may be due to maintaining native conditions during our MS analysis which seems to preserve the strong binding affinity of cTnI to TnC. To preserve ejected subunit integrity, we needed to avoid excessive energetic activation which would lead to subunit fragmentation instead of just subunit ejection. Without access to an additional isolation or MS3 step on the FTICR-MS, we ran out of additional options to further activate the cTn(I-C) heterodimer without producing subunit fragmentation events. Additionally, it is also possible that the ejected cTnI monomer may appear at too low of a charge (*i.e.* too high of m/z range) for us to meaningfully detect during our complex-up native MS analysis on our FTICR-MS, which is not optimized at very high m/z . Critically, we found that the cTn(I-C) dimer was difficult to fragment by both CAD and ECD, which supports the strong binding affinity of cTnI to TnC. An extremely high collision voltage (70 V) was needed to generate fragments that were representative of cTnI monomer upon isolating the cTn(I-C) dimer for complex-down analysis. As shown in Supplementary Figure 10, we were only able to confirm cTnI monomer fragments in the N- and C- termini regions likely due to the external regions being solvent-exposed and internal regions of cTnI being in complex with TnC.

5. Given that cTnI is not detected, nor do the stripped complexes, the validity of the calculation in Supplementary Figure 8b is not clear.

Response: We thank the Reviewer for this critical comment. After careful consideration, we have removed Supplementary Figure 8b to avoid confusion.

6. Figure 3, the dashed lines that connect the insets to the figure is not clear and confusing.

Response: We thank the Reviewer for raising this concern. We have now changed Figure 3 to exclude the dashed lines that connect the insets to the figure. Additionally, we have removed the insets and changed them to their own separate Figure panels. We have updated the Figure caption to reflect this change as well.

Figure 3. Native top-down MS (nTDMS) analysis for the characterization of cTn complexes directly from human heart tissue. (a-f). (a) Isolation of the heterotrimeric cTn complex achieved by ultrahigh-resolution FTICR-MS with charge states labeled. Inset shows three common approaches used for nTDMS analysis. (b) Complex-up analysis of the cTn complex ejects cTn monomers and dimer to reveal heterotrimeric complex stoichiometry. (c) Isolation of ejected cTnT native monomer ($z = 14+$) followed by complex-down analysis using collisionally activated dissociation (CAD) fragmentation. Theoretical isotope distributions (red circles) are overlaid on

the experimentally obtained mass spectrum to illustrate the high mass accuracy. All individual ion assignments are < 2 ppm from the theoretical mass. (d) Representative MS/MS spectra and CAD fragment ions ($y_{17}^{2+}, y_{39}^{4+}, y_{118}^{9+}$) obtained from the nTDMS analysis. (e) Isolation of ejected cTn native dimer ($z = 14+$) followed by complex-down analysis using CAD fragmentation. Theoretical isotope distributions (red circles) are overlaid on the experimentally obtained mass spectrum to illustrate the high mass accuracy. All individual ion assignments are within 2 ppm from the theoretical mass. (f) Representative MS/MS spectra and CAD fragment ions (cTnI: y_{42}^{3+}, b_{26}^{1+} , TnC: y_{16}^{1+}, y_{20}^{1+}) obtained from the nTDMS analysis.

7. On page 13, the authors discuss "collisional activation ramping" experiments aimed at identifying the "least" and "most" Ca²⁺ binding region. However, the basis for their conclusion regarding the order of sequential binding of calcium ions remains unclear. It is important to understand why they assume that sensitivity to collision activation correlates with the binding order of calcium ions.

Response: We thank the Reviewer for the excellent comment. To clarify, we had some prior knowledge regarding the potential region and order of sequential binding of Ca²⁺ ions based on the UniprotKB database (P63316) and previous publications using recombinant TnC and/or cardiomyocytes.¹²⁻¹⁶ In TnC Ca²⁺ binds to domain II with low-affinity ($\sim 10^{-5}$ M) and is known as the “regulatory” domain that is important for initiating cardiac contraction.¹⁴ During relaxation (low intracellular Ca²⁺ concentrations), domain II is rarely occupied, however during cardiac contraction (high intracellular Ca²⁺ concentration) domain II becomes significantly more occupied. On the other hand, domains III and IV are known as “structural” sites that can bind Ca²⁺ with high-affinity ($\sim 10^7$ M⁻¹).¹⁵ Both domains are typically saturated with Ca²⁺ during relaxation and contraction.¹⁶ Additionally, it has also been found that domains III and IV have a slower Ca²⁺ exchange rate than domain II.^{13, 15} Although previous investigations employing recombinant TnC and cardiomyocytes have contributed valuable insights into TnC Ca²⁺ sensitivity, these studies have often focused on isolated domains or the cTn(I-C) heterodimer, and have not conducted direct examinations of the whole cTn(I-T-C) complex. In contrast, our nTDMS results offer indispensable structural insights, particularly through our collisional activation studies of the intact endogenous cTn(I-T-C) complex. We have now added in this additional discussion to address the Reviewers comments on Pages 13, 20, and 21.

During our nTDMS collisional activation experiments we directly probed what the most resilient Ca²⁺ binding domains are in TnC monomer. Our experiments inform us of the relative stability of the various Ca²⁺ binding domains and can reveal what the preferred Ca²⁺ binding sites in TnC are. For example, when isolating TnC + 1 Ca²⁺ (2312 m/z), we did not observe any sequence informative fragment ions related to domain IV. Only when we isolated TnC + 2 Ca²⁺ (2316 m/z) did we see sequence informative fragment ions for domains III and IV. Similarly, when we isolated TnC + 3 Ca²⁺ (2321 m/z) we observed sequence informative fragments for domains III, IV, and II. If there was no bias for the primary Ca²⁺ bound state, then we would expect to observe a stochastic distribution of Ca²⁺ domains for only the 1 Ca²⁺ bound species. However, our collisional activation experiments reveal that TnC Ca²⁺ binding is not stochastic and instead shows intrinsic bias to specific Ca²⁺ binding regions. In all cases, we can infer that the relative binding order suggested

by our collisional activation experiments reveals a preferred Ca^{2+} sensitivity order. Therefore, we believe that Ca^{2+} binding to domain III, IV and II is a sequential event due to this observation during the nTDMS experiments. Overall, our nTDMS results are highly informative and provide new insights to regional Ca^{2+} binding sensitivity of the whole endogenous cTn(I-T-C) heterotrimer complex.

8. *From the top-down data how were the 3-5 amino acids domains of calcium binding defined, given that the fragmentation map indicates longer peptides?*

Response: We thank the Reviewer for the question. For clarification, the domains we highlighted were determined through native top-down MS/MS analysis to be the essential amino acid regions necessary for an individual calcium ion to associate. Aspartic acid and glutamic acid tend to have the highest binding affinities for calcium ions at neutral pH.^{12, 17} In the present case, all fragment ions containing both Asp and/or Glu residues on either side of the highlighted regions presented with a bound calcium ion and all fragment ions localized with only one of the Asp or Glu residues on either side were found to have lost calcium ion association. We note in the text (page 21) that there have been previous reports of possible TnC calcium binding sites listed on the Uniprot database that our results further refine. We manually validated our top-down data using MASH Native¹⁸ by adding the monoisotopic mass of calcium to every possible amino acid residue in the potential calcium binding regions listed on Uniprot. We have added a clarification sentence in the main text describing how we determined the 3-5 amino acid domains of calcium binding in TnC on page 13.

9. *The data of the “collisional activation ramping” experiments is not shown. Only a single spectrum.*

Response: We thank the Reviewer for the excellent comment. We have added a new Supplementary Figure 12 that shows the collisional activation ramping experiments we performed on TnC monomer.

Figure S12. Collisional activation ramping of TnC monomer by native top-down MS. MS/MS characterization of the isolated TnC proteoform ($z = 8+$). A collisionally activated dissociation (CAD) energy of 4 V, 48 V, and 55 V were applied in the collision cell.

10. On page 17 it is indicated that “17 endogenous cTn proteoforms” were identified, however, only 10 cTn(I-T-C) proteoforms are listed in Supplementary table 2.

Response: We thank the Reviewer for the thoughtful comment. The “17 endogenous cTn proteoforms” encompasses every cTn proteoform we detected from our native top-down MS experiments including endogenous cTn(I-T-C) heterotrimer, cTn(I-C) dimer, cTnT monomer, and TnC monomer proteoforms. For clarification, cTnT and TnC proteoforms are included in this count because we detected both monomers during our complex-up and complex-down native MS experiments. Hence, these monomers were ejected directly from the cTn(I-T-C) complex. We have now updated the title of Supplementary Table 2 to “cTn proteoforms identified in native top-down MS experiments”.

11. In supplementary Figure 2, TnC seems to be proteolytically degraded (batch 2 and 3), can the authors comment on this?

Response: We thank the Reviewer for the careful reading. TnC is not proteolytically degraded in batch 2 and 3 in Supplementary Figure 2. The bottom most bands in the E lanes of batch 2 and 3 correspond to some co-elution of myosin light chain 2-v (MLC-2v). While our NP-Pep platform can effectively deplete many highly abundant and high MW species present in the loading mixture, there is a small amount of MLC-2v left in the elution, which does not affect the cTn complex analysis.

12. How was the relative abundance in Supplementary Figure 4 calculated?

Response: We thank the Reviewer for the excellent question. We have corrected the y-axis in Supplementary Figure 4 (now Figure S5) to “Total Intensity”. The total intensity was calculated by summing the individual ion intensities of the various cTn proteoforms from the deconvoluted top-down mass spectra while normalizing for total protein amount (600 ng) injected between the loading mixture (L) and the elution mixture (E). Following the Reviewers comment, we have also revised the figure caption to clarify the calculation.

13. The entire scheme of OBE and the inclusion of panels b-d in Supplementary Figure 5 is unclear, as these results have already been published by the lab of Vicki Wysocki.

Response: We thank the Reviewer for raising these concerns. We have now excluded panels b-d in Supplementary Figure 5 (now Figure S6).

14. Figure 5, panels c and d - what do the dashed lines represent? What m/z range was isolated for the TIMS-MS experiments. Do they cover only one proteoform? How was the C-terminal lysine deletion in cTnT validated? What specie of the dimer was isolated for the validation of the double phosphorylation?

Response: We thank the Reviewer for the excellent questions. The dashed lines in the inset of Figure 5a and 5c highlight the different Ca²⁺ bound states in the ion mobility heat map for TnC and cTn(I-C) dimer for single charge state, respectively. The dashed lines in Figure 5b and 5d are drawn to illustrate changes to the CCS values between different Ca²⁺ bound states. To clarify, we did not isolate a specific m/z range for the TIMS-MS experiments and our TIMS-MS experiments cover all proteoforms observed for the specific protein species, as shown in Figure 5a and 5c. We were unable to quadrupole isolate any species above 3000 m/z due to instrumentation limitation of our timsTOF Pro mass spectrometer. We have corrected the caption in Figure 5 and removed the word “isolation” to more accurately reflect the data. Regarding proteoform identification, we had *a priori* knowledge about the specific proteoform identifies for the TIMS-MS analysis because of the previously obtained FTICR-MS/MS data. Regarding the C-terminal lysine deletion in cTnT, we manually validated this proteoform based on highly accurate intact mass measurements (within 1.2 ppm, shown in Figure 3c) together with the prior knowledge on identified cTnT proteoforms.¹⁹⁻²¹ For TnC, we detect 4 proteoforms of TnC in its different Ca²⁺ bound states. Whereas, for cTn(I-C) dimer (*ppcTnI* + TnC) we detect 2 proteoforms in its different Ca²⁺ bound states. Additionally, the double phosphorylated proteoforms for cTn(I-C) dimer were manually identified based on highly accurate intact mass measurements from our FTICR-MS/MS data.

15. In the ion mobility experiments in figure 5, how can the conditions for BSA analysis, a monomer without cofactors, assist in optimizing conditions for analysis of the complex, bound to Calcium?

Response: We thank the Reviewer for the constructive comment. We found BSA (~66 kDa) to be in a good model protein because it presents at a very similar m/z and ion mobility 1/K₀ to the cTn(I-T-C) complex. Being that our enrichments use precious human heart tissues, we are rather sample limited following elution of the cTn(I-T-C) complex. Thus, it was not optimal to use cTn to optimize the timsTOF method. BSA is inexpensive, was readily available, exists in a similar m/z

region, and similar ion mobility $1/K_0$ as the cTn(I-T-C) complex and was used as a model protein. To further clarify, we have now added a note on Page 16 as to why we chose BSA for optimizing our TIMS-MS analysis. This revision is recapitulated as follows: “First, the native TIMS-MS parameters were optimized using bovine serum albumin (BSA, ~ 66 kDa) due to BSA having similar m/z and ion mobility regions as the cTn complex.”

16. In figure 5, panels e and f are not required, they simply describe panels b and d in a different type of representation. Same goes for panel a in figure 6, which recapitulates panel e, and the structure of EGTA is completely irrelevant here. Also – what are the measured masses of the complexes in panel d? Can the authors deduce information on the level of calcium binding from the measured masses?

Response: We thank the Reviewer for these comments. The cartoon schematics in Figure 5e-f and Figure 6a are meant to guide a general audience for *Nature Communications* through the interpretation of the native TIMS-MS results. We feel it is important to leave these illustrations in Figures 5 and 6 because the interpretation of CCS values from the TIMS-MS data may be confusing to a non-MS specific audience. However, if both the Reviewers and the Editor think these illustrative panels should be removed, we are happy to oblige.

For Figure 6d, we did not achieve isotopic resolution of the cTn complexes. Therefore, the measured masses are not easily extracted from the data shown in Figure 6d as without isotopic resolution we can only achieve an average mass of the ensemble which is not entirely descriptive for such a heterogeneous protein complex such as the cTn complex. We also refrained from inferring the exact level of Ca^{2+} binding for this same reason. From the data shown in Figure 6, we prove that Ca^{2+} binding state is directly associated with cTn complex state, but it is difficult to show the specific forms present due to not having isotopic resolution of the individual proteoforms. It is critical to note that EGTA has a Ca^{2+} binding K_d ~61 nM at physiological pH. However, the association of cTnI with TnC tends to dramatically increase the sensitivity of TnC to Ca^{2+} binding. Davis *et al.* report that Ca^{2+} binding of TnC alone has ~ μM K_d values, however, with cTnI present it can be ~nM K_d values.²² Thus, there is a chance that we are not able to fully strip Ca^{2+} from the cTn complex because of this equilibrium of EGTA- Ca^{2+} K_d being within the same order as cTn- Ca^{2+} K_d . So, while we do strip off calcium from the cTn complex, we may not be able to do this completely and we suspect that the remaining cTn complex shown in Figure 6d still possesses some smaller amount of Ca^{2+} bound (the exact calcium binding number not clear). Following the Reviewers comments we have modified Figure 6a and have briefly explained in the main text why we cannot deduce the exact level of Ca^{2+} binding from the measured masses (Page 18).

Reviewer #2:

Chapman and coauthors developed a “native nanoproteomics” strategy for the enrichment of native cardiac troponin (cTn) complexes directly from human heart tissue using peptide-functionalized superparamagnetic nanoparticles under non-denaturing conditions, native top-down mass spectrometry (nTDMS) characterization of low abundance cTn complexes was

subsequently performed, which yields the stoichiometry and composition of the heterotrimeric cTn complex, localizes Ca²⁺ binding domains (II-IV), defines cTn-Ca²⁺ binding dynamics, and provides high-resolution mapping of the proteoform landscape.

Overall, this is an exciting work that opens a new paradigm for structural characterization of low-abundance native protein complexes. This manuscript is well written and the experimental is well designed. There are a few major concerns going back to the results as outlined below, along with some minor issues listed below that need to be addressed prior to publication:

Response: We thank the Reviewer for the highly positive comments on the manuscript and we greatly appreciate the Reviewer's questions and critiques. We have responded to the Reviewer's comments, which have improved the quality of our manuscript significantly.

1. When we look at Figure 2b, 2c-d, and Figure 3d together, there is something strange. As shown in Figure 2c and 2d, the most abundant cTn(I-T-C) complexes are cTnT(p)-cTnI(p)-TnC(2Ca²⁺), cTnT(p)-cTnI(p)-TnC(3Ca²⁺), cTnT(p)-cTnI(2p)-TnC(2Ca²⁺), and cTnT(p)-cTnI(2p)-TnC(3Ca²⁺); while Figure 3d displays that the ejected cTn(I-C) dimers are cTnI-TnC(2Ca²⁺), cTnI-TnC(3Ca²⁺), cTnI(p)-TnC(2Ca²⁺), cTnI(p)-TnC(3Ca²⁺), cTnI(2p)-TnC(2Ca²⁺), and cTnI(2p)-TnC(3Ca²⁺), among which, the ejected cTnI-TnC(2Ca²⁺) and cTnI-TnC(3Ca²⁺) dimers (the dimers with the wide type cTnI) are the most intensive ones, but their corresponding cTn(I-T-C) complexes (cTnT(p)-cTnI-TnC(2Ca²⁺) and cTnT(p)-cTnI-TnC(3Ca²⁺)) are of low abundance in Figure 2d (not labelled). Please address how this pheromone happened. Likewise, the abundance ratio for monomeric cTnI: cTnI(p) : cTnI(2p) is ~ 1:1:0.5 (Figure 2b); while when we sum the abundances of cTnI-TnC(2Ca²⁺) and cTnI-TnC(3Ca²⁺) for cTnI, cTnI(p)-TnC(2Ca²⁺) and cTnI(p)-TnC(3Ca²⁺) for cTnI(p), and cTnI(2p)-TnC(2Ca²⁺) + cTnI(2p)-TnC(3Ca²⁺) for cTnI(2p), respectively, the abundant ratio for cTnI: cTnI(p) : cTnI(2p) is 1:0.5:0.15 (Figure 3d); the abundant ratio for cTnI(p) : cTnI(2p) in Figure 2d is about 1:1 for cTnT(p)-cTnI(p)-TnC(2Ca²⁺) + cTnT(p)-cTnI(p)-TnC(3Ca²⁺), cTnT(p)-cTnI(2p)-TnC(2Ca²⁺) + cTnT(p)-cTnI(2p)-TnC(3Ca²⁺). please explain why would the abundance ratios of cTnI: cTnI(p) : cTnI(2p) at monomeric forms, cTn(I-C) dimers, and cTn(I-T-C) complexes vary so differently?

Response: We thank the Reviewer for the thoughtful questions. To clarify, the difference in abundance of phosphorylated cTnI forms between the monomer, dimers, and heterotrimers is due to biological variabilities in the proteoform landscape of different human donor hearts. For example, the cTn complexes shown in Figure 2c-d (now Figure 2a-b) are enriched from Donor 2 heart tissue (Table S1) which primarily consists of bisphosphorylated cTnI. However, the RPLC-MS data that was shown in Figure 2b (now Figure S3b) and cTn(I-C) dimer in Figure 3d is enriched from Donor 1 tissue which has various unphosphorylated, monophosphorylated, and bisphosphorylated cTnI forms. The biological variability of the human donor heart tissues has been reported in our previous publications.^{19, 20} We have now replaced the Figure 2b results (now Supplementary Figure 3b) to show the cTn proteoforms from Donor 1 to Donor 2 to avoid further confusion.

2. Page 13, in the following discussion, “Progressive collisional activation ramping revealed TnC domain III to be the least vulnerable Ca²⁺ binding to increasing collisional activation, while domain II was found to be the most vulnerable Ca²⁺ binding region. To localize the primary region for Ca²⁺ binding that is least vulnerable to collisional activation, we isolated the TnC proteoform at 2312 m/z and performed CAD to yield product ions y₅₂ + Ca²⁺, y₃₀, b₁₁₅ + Ca²⁺, and b₁₀₉ (Figure S12). Therefore, the primary Ca²⁺ binding domain was localized to 113DLD115 in domain III. The next Ca²⁺ binding domain was localized to the structural region between 141DKNND145 in domain IV by first isolating the TnC proteoform at 2316 m/z and then generating CAD product ions b₁₄₀, b₁₄₅ + Ca²⁺, y₁₆, and y₂₂ + Ca²⁺ (Figure S13). Finally, the most vulnerable Ca²⁺ binding region was localized to regulatory domain II between 73DFDE76 by isolating the TnC proteoform at 2321 m/z and generating CAD product ions b₆₅, b₉₁ + Ca²⁺, y₈₅, and y₉₄ + Ca²⁺ (Figure S14).”, it seems that the authors have some prior knowledge about the binding order of each Ca²⁺ ion, if yes, please clarify and add in corresponding references. Otherwise, it is rather strange to conclude that the binding strength can be related to specific domain through progressive collisional activation ramping. Additionally, the authors performed CAD experiments for TnC+1Ca at m/z 2312 (Figure S12), TnC +2Ca at m/z 2312 (Figure S13), and TnC +3Ca at m/z 2312 (Figure S14), respectively, and observed the primary Ca²⁺ binding domain as domain III. Did the authors see any Ca²⁺ binding fragments related to domain IV in Figure S12? If not, does this mean that the Ca²⁺ binding to domain IV is a sequential event upon the primary Ca²⁺ binding to domain III? Any biological significance? If the Ca²⁺ ion was found to bind to domain IV in CAD of TnC+1Ca, would that an indication of malfunction of TnC? Furthermore, for the CAD results for TnC+2Ca(II), it states “The next Ca²⁺ binding domain was localized to ... 141DKNND145 in domain IV ... generating CAD product ions b₁₄₀, b₁₄₅ + Ca²⁺, y₁₆, and y₂₂ + Ca²⁺ (Figure S13)”, the N-terminal product ions by AA140 should cover the primary Ca²⁺ binding site at domain III and AA145 should bind to the two Ca²⁺ ions bound to domain III and IV, something like, b₁₄₀ + Ca²⁺ and b₁₄₅ + 2Ca²⁺. Similarly, “Finally, the most vulnerable Ca²⁺ binding region was localized to regulatory domain II between 73DFDE76 by isolating the TnC proteoform at 2321 m/z and generating CAD product ions b₆₅, b₉₁ + Ca²⁺, y₈₅, and y₉₄ + Ca²⁺,” y₈₅ and y₉₄ should cover 2Ca²⁺ and 3 Ca²⁺ binding sites, respectively. Something like y₈₅+ 2Ca²⁺, and y₉₄+ 3Ca²⁺. Please replace the insert figures of these fragment ions with the ones that can simultaneously reflex other binding sites. Last but not least, there are mistakes in figure legends for Figure S12, S13, and S14. For example, it states “Representative collisionally activated dissociation (CAD) fragment ions (y₃₀ 2+, y₅₂ 2+, b₁₀₉ 6+, b₁₁₅ 6+)” in the manuscript, but in Figure S12(b), it says “Representative collisionally activated dissociation (CAD) fragment ions (y₃₀ 2+, y₅₂ 2+, b₁₀₉ 6+, b₁₁₅ 6+) obtained from the nTDMS analysis.”. The same mistakes also present in S13 and S14.

Response: We thank the Reviewer for the excellent comments and questions. To clarify we had some prior knowledge on what regions were most likely to bind Ca²⁺ in TnC based on the UniprotKB database (P63316) and previous publications.¹²⁻¹⁶ Ca²⁺ binds to domain II with low-affinity (~10⁻⁵ M) and is known as the “regulatory” domain that is important for initiating cardiac contraction.¹⁴ During relaxation (low intracellular Ca²⁺ concentrations), domain II is rarely occupied, however during cardiac contraction (high intracellular Ca²⁺ concentration) domain II becomes significantly more occupied. On the other hand, domains III and IV are known as

“structural” sites that can bind Ca^{2+} with high-affinity ($\sim 10^7 \text{ M}^{-1}$).¹⁵ Both domains are typically saturated with Ca^{2+} during relaxation and contraction.¹⁶ It has also been found that domains III and IV have a slower Ca^{2+} exchange rate than domain II.^{13, 15} It is important to note that the previously mentioned studies were carried out using recombinant TnC and/or skinned cardiomyocytes, which are model systems and do not directly describe endogenous TnC presentation. However, our data captures the exact Ca^{2+} binding states from endogenous cTn complex that has been purified directly from native human heart tissues. We have now clarified this in the main text and added the corresponding references on Page 13.

When isolating TnC + 1 Ca^{2+} (2312 m/z), we did not observe any sequence informative fragment ions related to domain IV. Only when we isolated TnC + 2 Ca (2316 m/z) did we see sequence informative fragment ions for domains III and IV. Similarly, when we isolated TnC + 3 Ca (2321 m/z) we observed sequence informative fragments for domains III, IV, and II. If there was no bias for the primary Ca^{2+} bound state, then we would expect to observe a stochastic distribution of Ca^{2+} domains for only the 1 Ca^{2+} bound species. However, we do not observe this. Instead, we observe what appears to be a preferred Ca^{2+} binding pattern for the 2 and 3 Ca^{2+} bound states. Therefore, as suggested by the Reviewer, we suspect that Ca^{2+} binding to domain IV and II occurs sequentially as states on Page 21. As mentioned above, TnC domains III and IV are “structural” domains that anchor TnC to the thin filament and are typically saturated with Ca^{2+} during cardiac relaxation and contraction, whereas domain II is a “regulatory” domain that regulates cardiac contraction and regulation through Ca^{2+} binding. Additionally, domains III and IV bind to Ca^{2+} with much higher affinity than domain II. Our data shows a similar pattern in which Ca^{2+} binding to domains III and IV is more stable than binding to domain II. A previous study by Negele *et al.* demonstrated that mutations within Ca^{2+} binding domains III and IV greatly reduced the affinity of TnC for the regulatory region of cTnI, thus affecting cardiac contraction.²³ Another study by Swindle *et al.* discovered mutations to domains III and IV during hypertrophic cardiomyopathy (HCM) significantly reduced the binding affinities of Ca^{2+} to TnC, indicating these domains may play a more important role in modulating cardiac contraction than what was previously known.²⁴ Therefore, perturbation in the affinity of TnC for Ca^{2+} can potentially lead to adverse physiological consequences, such as the development of cardiomyopathies. While we acknowledge the potential biological significance of the observed sequential binding event of Ca^{2+} to TnC in our data, our current understanding remains incomplete. The studies described above were not performed on endogenous TnC, but instead recombinant TnC and skinned muscle fibers. To gain a more comprehensive insight, we would need additional functional data of endogenous TnC to confirm our inferences. Finally, if the Ca^{2+} ion was found to bind to domain IV first, instead of III, there may not be a complete malfunction of TnC since it has been demonstrated that at least one active C-terminal domain is required for tight association of TnC to cTnI.²³ This requirement can be fulfilled by either domain III or IV. We have now added additional discussion on Page 21.

All fragment ions shown in Figure S13-15 reflect the appropriate binding domains for each isolation, the few sentences in the main text were incorrectly stated. We have now corrected the main text to reflect the correct information displayed in Supplementary Figures 13-15. We also corrected the Figure captions in Supplementary Figures 14 and 15 to represent the correct fragment

ions shown. Finally, we corrected the “collisionally” typo in the captions of Supplementary Figure 13 and 15. We are grateful to the Reviewer for the careful reading of our manuscript.

3. As shown in Table S1, the authors obtained and studied a list of 5 non-failing donor hearts from clinical donors with different causes of death, but in the manuscript, whether or not their disease conditions and causes of death affect the existing forms, stability, and dynamics of cTn complexes were not mentioned. It would be good to add in some discussion for this part.

Response: We thank the Reviewer for this excellent point. Our lab has previously demonstrated that phosphorylation of cTnI is a candidate biomarker for chronic heart failure.²⁵ Our previous studies have found that phosphorylated forms of cTnI exist in high abundance in non-failing donor hearts, whereas unphosphorylated forms of cTnI are in higher abundance in diseased hearts.^{19, 20, 26} In Table S1, we describe the non-failing donor hearts used in this study as all arising from “apparently healthy” clinical donors with no major heart diseases. The human cardiac tissue collection from non-failing donor hearts is described in further detail in the Methods section of the main text on pages 25-26. While phosphorylated cTnI is dominant across non-failing hearts, the relative abundance of the various phosphorylated cTnI forms (cTnI, *pcTnI*, and *ppcTnI*) do experience donor-to-donor variability, which has been reported in our previous publications.^{19, 20} Since this is not a clinical study, we are hesitant to make conclusions on the effect of the disease conditions and cause of death due to the small sample size, which is beyond the scope of this manuscript.

4. Page 4, “However, only partial structural information has been obtained from conventional X-ray crystallography excluding the intrinsically disordered but functionally critical regions of cTnI and cTnT. Moreover, the cTn structure is highly dynamic due to Ca²⁺ binding^{23, 27, 28} and PTMs^{9, 29, 30} that regulates muscle contraction, yet traditional methods have not effectively captured these dynamic conformational changes³¹. Furthermore, recombinantly expressed proteins have been used in previous studies thus important structural features vital to the function of the endogenous cTn complex within the sarcomere were lost^{32, 33}” The introduction about the structure information of cTn complex is a bit vague, it is better to specify what kind of information has been obtained by biophysical approaches such as X-ray, cryoEM, and what has been lost. It will in turn strength the significance of this work.

Response: We greatly appreciate the Reviewer’s excellent suggestion. Only partial structural information of the core domain of the human cTn complex in its Ca²⁺ saturated state has been obtained from conventional X-ray crystallography excluding the intrinsically disordered but functionally critical regions of the N- and C- terminal regions of both cTnI and cTnT.²⁷ For example, the N-terminal region of cTnI which contains the functionally critical phosphorylation sites at Ser22/23 is not present in the partial crystal structure. Additionally, the cryo-EM studies are only able visualize the cTn complex on the thin filament in a static state. Being that the complex is a highly dynamic entity, methodologies for measuring cTn complex dynamics upon Ca²⁺ binding are needed. We have revised the introduction and added more information about the structure of the cTn complex in the introduction on Pages 4-5.

5. page 9, in Figure 2d, for the zoomed-in mass spectra from c is within m/z 4057 – 4067; therefore, the dash line displays the zoomed-in region should be corrected accordingly.

Response: We thank the Reviewer for this important point. We have now corrected the dashed lines in Figure 2d (now Figure 2b) to show the accurate zoomed-in region within m/z 4057-4067.

6. page 10, “In-depth examination of the endogenous cTn complex revealed four unique proteoforms comprised of both covalent and non-covalent modifications (Figure 2d)” should be “... four unique proteoform complexes ...”.

Response: We thank the Reviewer for their suggestion. We have now corrected the sentence on page 10 to “In-depth examination of the endogenous cTn complex revealed four unique complex proteoforms comprised of both covalent and non-covalent modifications (**Figure 2b**).”

7. page 10, “mono- and bis-phosphorylated cTnI, and TnC with three bound Ca^{2+} ions (most abundant cTn complex $MW = 77136$ Da)” should be “mono-phosphorylated cTnI”.

Response: We thank the Reviewer for their suggestion. We have now corrected the sentence on page 10 to include “mono-phosphorylated cTnI”.

8. Page 11, “Our nTDMS analysis also suggests that both the intrinsically disordered C- and N-termini of cTnI are more solvent exposed than the stable internal regions that form the subunit-subunit interfaces of the cTn complex.”, corresponding reference(s) should be added.

Response: We greatly appreciate the suggestion. We have now added the appropriate reference to the sentence on Page 11.

9. Page 15, “Due to the conformational heterogeneity and presence of intrinsically disordered regions along the heterotrimeric cTn complex, it is challenging to obtain crystal structures of cTn in its active and closed states upon Ca^{2+} binding using traditional structural biology techniques⁴³. Please specify what does the “closed states” exactly mean?”

Response: We thank the Reviewer for this question. To clarify, the cTn complex is a highly dynamic molecule that easily flips between relaxed and active conformations to perform its role in regulating cardiac contraction in response to intracellular Ca^{2+} concentration. During cardiac contraction, the core of the heterotrimeric cTn complex maintains a stable conformation, while flexible regions undergo extensive conformational changes when Ca^{2+} binds to TnC from the so-called ‘closed’ state to an active ‘open state’ in a “hinge-like motion”.²⁸ For example, the “closed state” of the cTn complex refers to when there are low intracellular levels of Ca^{2+} and the flexible cTnI “switch peptide” region is not able to bind to the N-terminal domain of TnC within the complex. Instead, cTnI prevents actin-myosin cross bridge formation i.e. cardiac muscle contraction by binding to actin and locking tropomyosin into a fixed position on the thin filament. When intracellular levels of Ca^{2+} increase and the cTn complex moves into its “open state”, Ca^{2+} binds to TnC and induces significant conformational changes within the cTn complex. The N-terminal domain of TnC opens for binding to cTnI, thus permitting actin to freely bind to myosin

for cardiac muscle contraction. We have added the following sentence to Page 15: “The core of the heterotrimeric cTn complex maintains a stable conformation, while flexible regions undergo extensive conformational changes when Ca^{2+} binds to TnC. This transition shifts the cTn complex from a ‘closed’ state where muscle contraction is prevented by cTnI binding to actin, to an active ‘open state’. In this active state, the N-terminal domain of TnC opens, allowing for the binding of cTnI, and facilitating the interaction between actin and myosin, ultimately leading to cardiac muscle contraction.”

10. Page 15, “The CCS values for TnC monomer ... for the most abundant charge state ($z = 8+$), respectively (Figure 5a-b).”, it should be $z=7+$. Similarly, “On the other hand, the CCS values for cTn(I-C) dimer with 2 (3623 \AA^2) and 3 (3640 \AA^2) Ca^{2+} ions for the most abundant charge state ($z = 15+$) ... (Figure 5c-d, Table S3)”, it should be $z=13+$.

Response: We thank the Reviewer for their careful reading of our manuscript. We have corrected the typos in Figure 5 to reflect the correct charge states shown for TnC ($z = 7+$) and cTn(I-C) dimer ($z = 13+$).

11. Page 17, “conformation to prepare the N-terminal region of TnC for binding to the C-terminal region of cTnI to initiate cardiac muscle contraction. Corresponding reference(s) should be added.

Response: We appreciate the Reviewer for their suggestion. We have added the appropriate references for the above sentence on Page 17.

Reviewer #3

Thank you for the comprehensive description of the antibody-free top-down approach to specifically enrich the low-abundance troponin complex. The manuscript is very well written; clear, precise, and easy to understand. Some text is repetitive from the previous publication <https://doi.org/10.1038/s41467-020-17643-1> but I understand that the point has to be made. The following minor suggestions are:

Response: We thank the Reviewer for the highly positive comments and approval of our work. We have addressed the remaining minor comments below.

1. *There method has been developed and described by the group before, please rewrite the sentence at page 4: “Here, we have developed a “native nanoproteomics” platform integrating the native enrichment of low-abundance protein complexes directly from tissues using surface functionalized superparamagnetic nanoparticles (NPs) with high-resolution nTDMS to characterize the structure and dynamics of low-abundance endogenous protein complexes for the first time”.*

Author used different name to describe the top-down method, current manuscript: native top-down mass spectrometry (nTDMS) compared to previous manuscript: top-down LC/MS coupling reversed-phase liquid chromatography (RPLC) to high-resolution MS. However, based on the method section both manuscripts used the same instruments and set up to run the samples for the general cTn identification. Therefore, I encourage author to acknowledge previous publication. I

do understand that previous manuscript concentrated on TnI however as described and presented, for example in supplement figure 4 and 7 the whole cTn complex was enriched and the peptide used to functionalize the NP surface was the same for both experiments/manuscripts.

Response: We thank the Reviewer for this constructive comment. To clarify, although our native nanoproteomics platform uses the same peptide-functionalized nanoparticle design as in our previous work, the native extraction method from human tissues, native enrichment strategy, and native top-down MS techniques presented in this manuscript are novel and were not previously described. As the Reviewer correctly summarized, our previous work focused solely on the enrichment of cTnI proteoforms from human tissue and serum and subsequent top-down RPLC-MS analysis under *denaturing* conditions. In our current manuscript, we have devised a new strategy to purify and analyze endogenous cTn(I-T-C) complex under *non-denaturing* conditions. While there are sections in the present manuscript that used similar setups as in our previous publication for the denaturing RPLC-MS analysis, the methods described in our previous manuscript would not be suitable for native cTn complex purification and native top-down MS analysis. We agree with the Reviewer's suggestion to acknowledge our previous publication more clearly, and, to this end, we have cited our previous publication when appropriate on pages 5, 8, 24, and 25 in the main text and on pages S-5, S-6 and S-11 in the supplementary information.

2. Did author measure the cTn percent recovery of the NP-Pep?

Response: We thank the Reviewer for the excellent suggestion. To find the cTn percent recovery, we measured the concentration of cTnI in SDS-PAGE gel bands between the loading mixture and elution mixture using ImageJ software. The concentration of cTnI was used to calculate the cTn percent recovery as opposed to TnC and cTnT because the peptide only has high affinity towards cTnI. We measured that cTnI had an 89% recovery in the elution mixture. The recovery percentage was calculated as follows: % Recovery = [the amount of cTnI in the elution (after enrichment) / the original amount of cTnI in the loading mixture (before enrichment)] x 100%.²⁹ Following the Reviewers comment, we have included the measured cTnI percent recovery in Figure S3.

3. Grammar: page 3. "These present tremendous challenges to studying their structure and dynamics using..." should be "to study"

Response: We thank the Reviewer for the careful reading. We have now changed the sentence on page 3 to "These present tremendous challenges to study their structure and dynamics using...".

4. Please rewrite the sentence, it does not make sense, page 4: "Furthermore, recombinantly expressed proteins have been used in previous studies thus important structural features vital to the function of the endogenous cTn complex within the sarcomere were lost".

Response: We thank the Reviewer for the careful reading. Following the Reviewers comment, we have now rewritten the sentence now on page 5.

5. Please rewrite the sentence, as verb is missing; figure 4 : ” (d) Structural representation of the cTn complex with the three Ca(II)-binding domains (II-IV) discovered in these experiments highlighted (domain II, blue; domain III, red; domain IV, yellow, UniprotKB annotations, gray).

Response: We thank the Reviewer for the careful reading. We have rewritten the sentence in the figure caption of Figure 4 on page 15. The sentence is rewritten as following “Structural representation of the cTn complex with the experimentally defined Ca(II)-binding domains (II-IV) highlighted (domain II, blue; domain III, red; domain IV, yellow, UniprotKB annotations, gray)”.

6. Please avoid text repetition, e.g. introduction creeping into the discussion part: “investigated by X-ray, crystallography, NMR, and cryo-EM”, page 3, 4, and 19. The problem was described in the introduction, it does not have to be repeated one more time.

Response: We thank the Reviewer for the constructive comment. We have now omitted the repetition of “X-ray crystallography, NMS, and cryo-EM” on page 20 in the discussion.

7. In the discussion maybe author can compare the phosphorylation analysis between current NP results to results from affinity purification performed by the same group (Tiambeng TN, Tucholski T, Wu Z, Zhu Y, Mitchell SD, Roberts DS, Jin Y, Ge Y. Analysis of cardiac troponin proteoforms by top-down mass spectrometry. *Methods Enzymol.* 2019;626:347-374. doi:10.1016/bs.mie.2019.07.029).

Response: We thank the Reviewer for their excellent suggestion. The phosphorylation analysis performed in our previous publication (doi:10.1016/bs.mie.2019.07.029) performed affinity purification of cTn with swine tissues. Additionally, following the affinity purification, online RPLC-MS/MS and offline MS/MS in *denaturing* conditions were used to evaluate swine cTn phosphorylation. In our current work we used *native* top-down MS strategies to evaluate phosphorylation and Ca²⁺ binding to cTn heterotrimers, dimers, and monomers directly from human tissues. Moreover, our native nanoproteomics method provides additional context about the diverse cTn complex proteoforms that our previous method using affinity purification was not able to illuminate.

8. Author uses PTMs but described only phosphorylation. Please keep in mind other modifications, including acetylation, methylation, oxidation, among others were reported on the troponin 's.

Response: We thank the Reviewer for the insightful comment. To clarify, we have reported in Supplementary Table 2 the breadth of various PTMs we detected (N-terminal acetylation, phosphorylation, and methionine (Met) removal). We did not detect methylation of troponin. To address this point, we have revised a sentence in the discussion section (page 21) to include the various PTMs reported in our study.

References

1. Yen, H.-Y. et al. Mass spectrometry captures biased signalling and allosteric modulation of a G-protein-coupled receptor. *Nature Chemistry* **14**, 1375-1382 (2022).
2. Wohlschlager, T. et al. Native mass spectrometry combined with enzymatic dissection unravels glycoform heterogeneity of biopharmaceuticals. *Nature Communications* **9**, 1713 (2018).
3. Chen, S. et al. Capturing a rhodopsin receptor signalling cascade across a native membrane. *Nature* **604**, 384-390 (2022).
4. Black, K.A. et al. A constricted opening in Kir channels does not impede potassium conduction. *Nature Communications* **11**, 3024 (2020).
5. Yen, H.-Y. et al. Ligand binding to a G protein-coupled receptor captured in a mass spectrometer. *Science Advances* **3**, e1701016.
6. Lermyte, F., Tsybin, Y.O., O'Connor, P.B. & Loo, J.A. Top or Middle? Up or Down? Toward a Standard Lexicon for Protein Top-Down and Allied Mass Spectrometry Approaches. *Journal of the American Society for Mass Spectrometry* **30**, 1149-1157 (2019).
7. Esteban, O. et al. Stoichiometry and Localization of the Stator Subunits E and Gin Thermus thermophilus H⁺-ATPase/Synthase*. *Journal of Biological Chemistry* **283**, 2595-2603 (2008).
8. Tucholski, T. et al. A Top-Down Proteomics Platform Coupling Serial Size Exclusion Chromatography and Fourier Transform Ion Cyclotron Resonance Mass Spectrometry. *Analytical Chemistry* **91**, 3835-3844 (2019).
9. Hwang, P.M., Cai, F., Pineda-Sanabria, S.E., Corson, D.C. & Sykes, B.D. The cardiac-specific N-terminal region of troponin I positions the regulatory domain of troponin C. *Proceedings of the National Academy of Sciences* **111**, 14412-14417 (2014).
10. Ferrières, G. et al. Systematic mapping of regions of human cardiac troponin I involved in binding to cardiac troponin C: N- and C-terminal low affinity contributing regions. *FEBS Letters* **479**, 99-105 (2000).
11. Park, K.C., Gaze, D.C., Collinson, P.O. & Marber, M.S. Cardiac troponins: from myocardial infarction to chronic disease. *Cardiovascular Research* **113**, 1708-1718 (2017).
12. Rayani, K. et al. Binding of calcium and magnesium to human cardiac troponin C. *Journal of Biological Chemistry* **296**, 100350 (2021).
13. Johnson, J.D., Nakkula, R.J., Vasulka, C. & Smillie, L.B. Modulation of Ca²⁺ exchange with the Ca(2+)-specific regulatory sites of troponin C. *Journal of Biological Chemistry* **269**, 8919-8923 (1994).
14. Cheung, J.Y., Tillotson, D.L., Yelamarty, R.V. & Scaduto, R.C. Cytosolic free calcium concentration in individual cardiac myocytes in primary culture. *American Journal of Physiology-Cell Physiology* **256**, C1120-C1130 (1989).
15. Schober, T. et al. Myofilament Ca Sensitization Increases Cytosolic Ca Binding Affinity, Alters Intracellular Ca Homeostasis, and Causes Pause-Dependent Ca-Triggered Arrhythmia. *Circulation Research* **111**, 170-179 (2012).
16. Bers, D.M. Calcium Fluxes Involved in Control of Cardiac Myocyte Contraction. *Circulation Research* **87**, 275-281 (2000).
17. Tang, N. & Skibsted, L.H. Calcium Binding to Amino Acids and Small Glycine Peptides in Aqueous Solution: Toward Peptide Design for Better Calcium Bioavailability. *Journal of Agricultural and Food Chemistry* **64**, 4376-4389 (2016).
18. Larson, E.J. et al. MASH Native: A Unified Solution for Native Top-Down Proteomics Data Processing. *Bioinformatics*, btad359 (2023).
19. Tucholski, T. et al. Distinct hypertrophic cardiomyopathy genotypes result in convergent sarcomeric proteoform profiles revealed by top-down proteomics. *Proceedings of the National Academy of Sciences* **117**, 24691-24700 (2020).
20. Chapman, E.A. et al. Defining the Sarcomeric Proteoform Landscape in Ischemic Cardiomyopathy by Top-Down Proteomics. *Journal of Proteome Research* **22**, 931-941 (2023).

21. Zhang, J. et al. Phosphorylation, but Not Alternative Splicing or Proteolytic Degradation, Is Conserved in Human and Mouse Cardiac Troponin T. *Biochemistry* **50**, 6081-6092 (2011).
22. Little, S., Tikunova, S., Swartz, D., Norman, C. & Davis, J. Measurement of Calcium Dissociation Rates from Troponin C in Rigor Skeletal Myofibrils. *Frontiers in Physiology* **2** (2011).
23. Negele, J.C., Dotson, D.G., Liu, W., Sweeney, H.L. & Putkey, J.A. Mutation of the high affinity calcium binding sites in cardiac troponin C. *Journal of Biological Chemistry* **267**, 825-831 (1992).
24. Swindle, N. & Tikunova, S.B. Hypertrophic Cardiomyopathy-Linked Mutation D145E Drastically Alters Calcium Binding by the C-Domain of Cardiac Troponin C. *Biochemistry* **49**, 4813-4820 (2010).
25. Zhang, J. et al. Top-Down Quantitative Proteomics Identified Phosphorylation of Cardiac Troponin I as a Candidate Biomarker for Chronic Heart Failure. *Journal of Proteome Research* **10**, 4054-4065 (2011).
26. Tiambeng, T.N. et al. Nanoproteomics enables proteoform-resolved analysis of low-abundance proteins in human serum. *Nature Communications* **11**, 3903 (2020).
27. Takeda, S., Yamashita, A., Maeda, K. & Maéda, Y. Structure of the core domain of human cardiac troponin in the Ca²⁺-saturated form. *Nature* **424**, 35-41 (2003).
28. Marston, S. & Zamora, J.E. Troponin structure and function: a view of recent progress. *Journal of Muscle Research and Cell Motility* **41**, 71-89 (2020).
29. Hwang, L. et al. Specific Enrichment of Phosphoproteins Using Functionalized Multivalent Nanoparticles. *Journal of the American Chemical Society* **137**, 2432-2435 (2015).

REVIEWERS' COMMENTS

Reviewer #1 (Remarks to the Author):

I continue to observe a significant disparity between the manuscript's title and abstract when compared to the actual study. This is because the method is exclusively elaborated and illustrated in the context of a single protein complex. Therefore, both the title and abstract should be revised. Any potential for future application of the method to multiple protein complexes should be reserved for the discussion section."

While the authors have addressed many points raised during the review by providing explanations, they should incorporate these descriptions into the manuscript to enhance the readers' comprehension of the study. For example:

- The title and abstract should be rephrased.
- The fact that "multiple depletions of highly abundant proteins" was done prior to the mass measurements shown in Fig. 2, should be mentioned in the results section.
- The term "low-abundance" should be replaced in Fig. 1 legend title of and in the discussion.
- The reason why "stripped complexes" are not detected should be included in the text.
- The fact that "FTICR could not infer exactly how Ca²⁺ binding and phosphorylation impacted the stability of the protein complexes" should be mentioned.
- A comment why cTnI is not stripped from the complex during MS/MS should be included.
- The word "isolation" should be removed from the legend of fig. 6

Reviewer #2 (Remarks to the Author):

The Authors have addressed all of my concerns with the original manuscript. The revised manuscript is ready for publication.

Responses to Reviewer Comments

Reviewer #1:

I continue to observe a significant disparity between the manuscript's title and abstract when compared to the actual study. This is because the method is exclusively elaborated and illustrated in the context of a single protein complex. Therefore, both the title and abstract should be revised. Any potential for future application of the method to multiple protein complexes should be reserved for the discussion section."

While the authors have addressed many points raised during the review by providing explanations, they should incorporate these descriptions into the manuscript to enhance the readers' comprehension of the study.

Response: We thank the Reviewer for the comments on the manuscript and we greatly appreciate the Reviewer's critiques. We have addressed the remaining minor comments below.

The title and abstract should be rephrased.

Response: We thank the Reviewer for the comment. We have now revised the title of the manuscript to "Structure and dynamics of endogenous cardiac troponin complexes in human heart tissue captured by native nanoproteomics". Additionally, we have rephrased the abstract to more accurately reflect our study.

The fact that "multiple depletions of highly abundant proteins" was done prior to the mass measurements shown in Fig. 2, should be mentioned in the results section.

Response: We thank the Reviewer for the excellent suggestion. We have now added the fact that multiple depletions of highly abundant proteins were done prior to mass measurements in the results section on page 6.

The term "low-abundance" should be replaced in Fig. 1 legend title of and in the discussion.

Response: We thank the Reviewer for this critical comment. We have now replaced the term "low-abundance" in Figure 1 and in the discussion to "endogenous".

The reason why "stripped complexes" are not detected should be included in the text.

Response: We thank the Reviewer for their suggestion. We have now provided the reason why "stripped complexes" are not detected on pages 10 in the main text.

The fact that "FTICR could not infer exactly how Ca²⁺ binding and phosphorylation impacted the stability of the protein complexes" should be mentioned.

Response: We thank the Reviewer for the suggestion. We have now mentioned on page 10 in the main text the fact that FTICR could not infer exactly how Ca²⁺ binding and phosphorylation impacted the stability of the cTn complexes.

A comment why cTnI is not stripped from the complex during MS/MS should be included.

Response: We thank the Reviewer for the excellent suggestion. We have now included a comment on page 9 explaining why we do not observe stripped cTnI monomer during our complex-up analysis.

The word “isolation” should be removed from the legend of fig. 6.

Response: We thank the Reviewer for the careful reading. We have now removed the word “isolation” from the legend of Figure 6.

Reviewer #2:

The Authors have addressed all of my concerns with the original manuscript. The revised manuscript is ready for publication.

Response: We thank the Reviewer for the positive comments and approval of our work.